# Unbiased Dynamic Pruning for Efficient Group-Based Policy Optimization

Haodong Zhu [* 1 2]   Yangyang Ren [* 1 2]   Yanjing Li [1]   Mingbao Lin [3]   Linlin Yang [4]
Xuhui Liu [5]   Xiantong Zhen [6]   Haiguang Liu [2]   Baochang Zhang [† 1]

## Abstract

Group Relative Policy Optimization (GRPO) effectively scales LLM reasoning but incurs prohibitive computational costs due to its extensive group-based sampling requirement. While recent selective data utilization methods can mitigate this overhead, they could induce estimation bias by altering the underlying sampling distribution, compromising theoretical rigor and convergence behavior. To address this limitation, we propose Dynamic Pruning Policy Optimization (DPPO), a framework that enables dynamic pruning while preserving unbiased gradient estimation through importance sampling-based correction. By incorporating mathematically derived rescaling factors, DPPO significantly accelerates GRPO training without altering the optimization objective of the full-batch baseline. Furthermore, to mitigate the data sparsity induced by pruning, we introduce Dense Prompt Packing, a window-based greedy strategy that maximizes valid token density and hardware utilization. Extensive experiments demonstrate that DPPO consistently accelerates training across diverse models and benchmarks. For instance, on Qwen3-4B trained on MATH, DPPO achieves $2.37\times$ training speedup and outperforms GRPO by 3.36% in average accuracy across six mathematical reasoning benchmarks.

## 1. Introduction

Reinforcement learning (RL) has emerged as a pivotal paradigm for scaling language models (Jaech et al., 2024; Guo et al., 2025; Yang et al., 2025). The use of large-

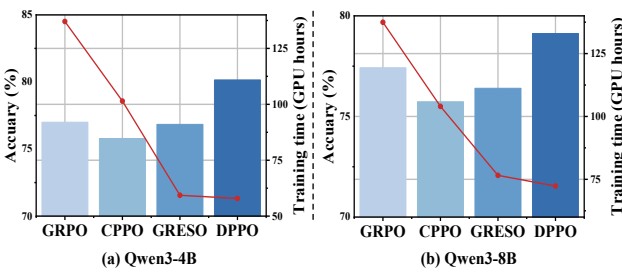

*Figure 1.* Comparison of accuracy and training time on the MATH dataset for Qwen3-4B and Qwen3-8B. Bars indicate accuracy and red lines indicate training time. Our method achieves the highest accuracy while requiring the least training time.

scale RL enables models to tackle increasingly sophisticated problems, such as competition-level mathematics and programming, through deeper and longer-horizon reasoning. A representative advancement in this direction is Group Relative Policy Optimization (GRPO), introduced by DeepSeek-R1 (Guo et al., 2025). In contrast to standard Proximal Policy Optimization (PPO) (Schulman et al., 2017), GRPO removes the value function critic and instead derives the baseline directly from group-level scores.

Despite its effectiveness, GRPO incurs substantial computational overhead. Specifically, it requires sampling a group of completions for each prompt in order to estimate intra-group advantages, causing the forward-pass cost to scale linearly with the group size. This burden is further amplified by the need to evaluate rule-based rewards and relative advantages for each completion. Moreover, to ensure training stability, GRPO computes probability ratios across the current, reference, and old policies, which significantly exacerbates the overall training cost. Together, these factors constrain the scalability and efficiency of GRPO, making training acceleration a critical research challenge.

To alleviate these computational bottlenecks, recent studies have explored selective data utilization strategies, which can be broadly categorized into prompt-level selection (*e.g.*, GRESO (Zheng et al., 2025b)) and completion-level selection (*e.g.*, CPPO (Lin et al., 2025)). While empirically effective in reducing training cost, these heuristic pruning approaches suffer from a fundamental limitation: **gradient bias**, as shown in Figure 2; that is, discarding samples deemed "low-value" alters the underlying sampling

---
[*]Equal contribution [†]Project lead. [1]Beihang University [2]Zhongguancun Academy [3]Rakuten Singapore [4]Communication University of China [5]KAUST [6]United Imaging Healthcare, Co., Ltd.. Correspondence to: Yanjing Li <yanjingli@buaa.edu.cn>, Linlin Yang <lyang@cuc.edu.cn>.

*Proceedings of the $43^{rd}$ International Conference on Machine Learning*, Seoul, South Korea. PMLR 306, 2026. Copyright 2026 by the author(s).

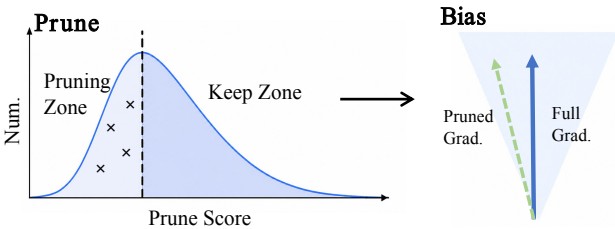

*Figure 2.* Selectively dropping training samples shifts the data distribution and biases the gradient. Existing methods discard low-scoring samples, whether prompts or completions, below a prune-score threshold (left), so the resulting gradient (Pruned Grad.) deviates from the full-data gradient (Full Grad.) (right).

distribution at both the prompt and completion levels, causing the resulting gradient estimates to systematically deviate from the true optimization objective (as analyzed in Sec. 4). Without appropriate theoretical correction, such bias causes suboptimal convergence and degraded policy performance, particularly in the sensitive optimization landscape of reasoning-oriented RL.

To bridge this gap, we propose **Dynamic Pruning Policy Optimization (DPPO)**, a dynamic pruning framework across both completion and prompt levels that reconciles computational efficiency with theoretical integrity. Unlike prior heuristic methods, DPPO employs a bias-corrected gradient formulation grounded in importance sampling principles. By weighting retained samples with mathematically derived rescaling factors, DPPO explicitly compensates for the distributional shift induced by dynamic pruning, ensuring that the expected gradient remains unbiased with respect to the full-batch baseline. Technically, DPPO implements a hierarchical, importance-aware pruning mechanism across both completion and prompt levels. At the completion level, it prunes responses with low information density to reduce backward-pass overhead, while at the prompt level, it filters redundant prompts to avoid wasteful rollouts.

In addition, selective pruning inevitably introduces data sparsity and fragmented memory access, which can undermine hardware utilization. To address this issue, we introduce **Dense Prompt Packing**, a window-based greedy strategy that reorganizes variable-length sequences into compact buffers. This design maximizes valid token density, improves hardware saturation, and mitigates memory fragmentation. Through the synergy of unbiased gradient correction and system-level optimization, DPPO achieves substantial training acceleration without compromising performance, as illustrated in Figure 1.

In summary, the primary contributions of this work are:

- We propose DPPO, an unbiased acceleration framework for GRPO based on hierarchical importance sampling. DPPO dynamically prunes redundancy at both

prompt and completion levels, overcoming the estimation bias inherent in prior heuristic approaches while preserving theoretical rigor.

- We introduce Dense Prompt Packing, a window-based greedy strategy that mitigates pruning-induced sparsity by efficiently organizing variable-length prompts, thereby sustaining high hardware throughput.

- Extensive experiments across multiple reasoning benchmarks demonstrate that DPPO significantly accelerates GRPO training while matching or surpassing the performance of the full-batch GRPO baseline.

**Conflict of Interest Disclosure** The authors declare no financial conflicts of interest.

## 2. Related Works

**RL Finetuning of LLMs.** Reinforcement learning (RL) has emerged as a cornerstone for aligning LLMs, initially demonstrating remarkable success through Reinforcement Learning with Human Feedback (RLHF) in enhancing instruction-following capabilities and safety (Christiano et al., 2017; Ouyang et al., 2022; Bai et al., 2022; Dong et al., 2024; Dai et al., 2024). More recently, the paradigm has shifted towards Reinforcement Learning with Verifiable Rewards (RLVR), which significantly improves reasoning capabilities in structured domains by leveraging automatically verifiable signals (Jaech et al., 2024; Guo et al., 2025; Team et al., 2025; Pan et al., 2025). In terms of policy optimization, while PPO (Schulman et al., 2017) remains widely adopted, Group Relative Policy Optimization (GRPO) (Guo et al., 2025) has gained substantial traction by eliminating the computationally expensive value network via group-normalized advantage estimation. To further enhance sample efficiency and stability, recent works have proposed targeted improvements to these algorithms (Liu et al., 2025b; Hu, 2025; Xiong et al., 2025). For instance, methods such as VinePPO (Kazemnejad et al., 2024) and VAPO (Yue et al., 2025) focus on refining value function optimization, whereas DAPO (Yu et al., 2025) specifically enhances GRPO through Dynamic Sampling, which filters zero-variance prompts to improve sample efficiency and training stability. Parallel efforts push the performance frontier across diverse model scales (Luo et al., 2025b;a; Zeng et al., 2025) and develop infrastructure for scalable RL training (Sheng et al., 2025).

**Data Selection for LLM.** Beyond algorithmic improvements, data selection has become pivotal for training efficiency. Initial approaches focused on supervised fine-tuning, demonstrating that pruning datasets to high-quality subsets can maintain performance (Xia et al., 2024; Ivison et al., 2025; Muennighoff et al., 2025; Chen et al., 2024). In the

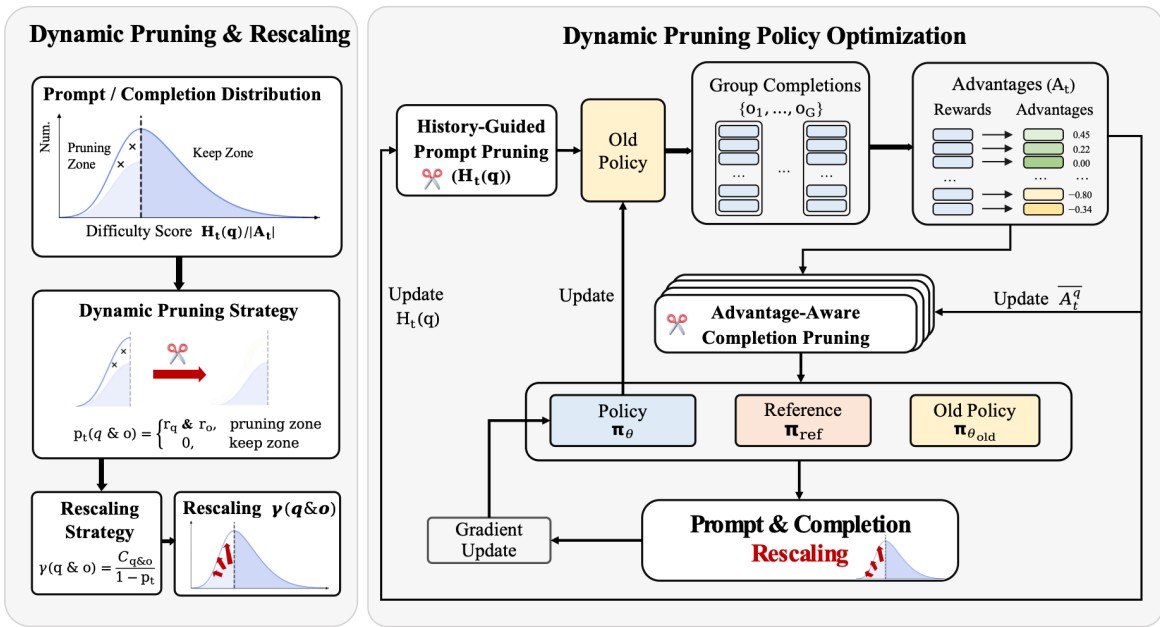

*Figure 3.* Overview of our DPPO. It employs a hierarchical pruning strategy to accelerate GRPO by reducing redundancy at both the prompt level (via difficulty estimation $H_t(q)$) and the completion level (via advantage assessment $|A_t|$). In the left panel, a mathematically grounded rescaling mechanism is applied to retained samples to correct for estimation bias. The right panel details the end-to-end training loop, where dynamic pruning and importance-based rescaling are integrated to ensure efficient yet unbiased policy optimization.

context of RL, data selection strategies generally fall into offline and online categories. Offline methods (Ye et al., 2025; Li et al., 2025; Wang et al., 2025; Fatemi et al., 2025; Muldrew et al., 2024) filter prompts prior to training based on static criteria. While showing that small subsets suffice for algorithms like GRPO (Li et al., 2025), they often incur pre-assessment overhead and fail to adapt to evolving training dynamics. Consequently, the focus has shifted to online selection, where methods dynamically filter prompts based on the current policy (Yu et al., 2025; Zhang et al., 2025). Approaches such as Dynamic Sampling (Yu et al., 2025) and others (Bae et al., 2025; Liu et al., 2025a; Cui et al., 2025; Meng et al., 2025) perform per-step filtering to remove ineffective samples, or employ curriculum-based strategies (Zheng et al., 2025b; Chen et al., 2025). However, these methods typically rely on heuristic pruning that alters the data distribution without theoretical correction, thereby introducing estimation bias, or incur substantial computational costs due to extra rollout evaluations. By reformulating pruning as an importance sampling process, our DPPO reweights retained samples to correct for distributional shifts, achieving significant acceleration while preserving an unbiased gradient estimate.

## 3. Background: Group-Based Policy Optimization

Group Relative Policy Optimization (GRPO) (Guo et al., 2025) simplifies conventional Reinforcement Learning from

Human Feedback (RLHF) pipeline by eliminating the critic model and deriving the baseline directly from group-based scores. For a given query $q$ sampled from the dataset distribution $P(Q)$, GRPO generates a set of $G$ completions $\{o_1, o_2, \ldots, o_G\}$ using the behavior policy $\pi_{\theta_{old}}(\cdot|q)$. Let $\rho_{i,\tau}(\theta) = \frac{\pi_\theta(o_{i,t}|q, o_{i,<\tau})}{\pi_{\theta_{old}}(o_{i,t}|q, o_{i,<\tau})}$ denote the probability ratio. The policy $\pi_\theta$ is then optimized by maximizing the objective function:

$$\mathcal{J}(\theta) = \mathbb{E}_{q\sim P(Q),\{o_i\}_{i=1}^G \sim \pi_{\theta_{old}}(\cdot|q)} \left[ \frac{1}{G} \sum_{i=1}^G \left( \frac{1}{|o_i|} \sum_{\tau=1}^{|o_i|} \right. \right.$$
$$\min\left( \rho_{i,\tau}(\theta)A_i, \text{clip}(\rho_{i,\tau}(\theta), 1-\epsilon, 1+\epsilon)A_i \right)$$
$$\left. \left. - \beta \mathbb{D}_{KL}(\pi_\theta \| \pi_{ref}) \right) \right],$$
$$(1)$$

where the per-token KL divergence $\mathbb{D}_{KL}[\pi_\theta \| \pi_{ref}] = \hat{\rho} - \log \hat{\rho} - 1$, with $\hat{\rho} = \pi_{ref}(o_{i,\tau}|q, o_{i,<\tau})/\pi_\theta(o_{i,\tau}|q, o_{i,<\tau})$. The $\epsilon$ and $\beta$ are hyperparameters, while $\pi_{ref}$ denotes the reference model. The advantage $A_i$ is estimated by normalizing the rewards $\{r_1, r_2, \ldots, r_G\}$ within each group:

$$A_i = \frac{r_i - \text{mean}(\{r_1, r_2, \ldots, r_G\})}{\text{std}(\{r_1, r_2, \ldots, r_G\})}. \quad (2)$$

## 4. Method: Dynamic Pruning for Unbiased GRPO Acceleration

The primary computational bottleneck in Group Relative Policy Optimization (GRPO) stems from its reliance on

generating multiple completions per prompt to estimate intra-group advantages. As the group size increases, this requirement causes the forward-pass cost to scale linearly, resulting in substantial computational overhead. While data pruning appears to be a natural remedy, existing selective utilization strategies (Zheng et al., 2025b; Lin et al., 2025) inevitably introduce estimation bias by altering the underlying sampling distribution.

To address this challenge, we propose **Dynamic Pruning Policy Optimization (DPPO)**, a framework illustrated in Figure 3 that accelerates GRPO while preserving unbiased gradient estimation. DPPO achieves this through a hierarchical dynamic pruning mechanism operating at both the prompt and completion levels, coupled with explicit bias-correction via importance rescaling. Specifically, we define a completion-level pruning policy $\mathcal{P}_t^o$ and a prompt-level pruning policy $\mathcal{P}_t^q$, where $\mathcal{P}_t^o(o) \in [0, 1)$ and $\mathcal{P}_t^q(q) \in [0, 1)$ denote the pruning probabilities for completion $o$ and prompt $q$, respectively. Following these policies, we selectively prune low-value samples at both the completion and prompt levels. To ensure unbiased gradient estimation, retained samples are reweighted by rescaling factors derived via importance sampling, thereby preserving the expectation of the original gradient in a principled manner.

## 4.1. Unbiased Gradient Estimation under Hierarchical Pruning

We analyze DPPO from a theoretical aspect and show that its hierarchical pruning strategy yields an unbiased estimator of the original GRPO objective. The analysis by Hu et al. (2025) shows that removing the KL divergence constraint in maximizing $\mathcal{J}(\theta)$ does not degrade the model's core reasoning capabilities, as these are governed by the reward-aligned advantage term. Accordingly, we restrict our analysis to the optimization of the advantage component.

Since the clipping mechanism primarily serves as a safeguard against excessively large policy updates and remains largely inactive under appropriately chosen step sizes (Engstrom et al., 2020), we adopt the unclipped surrogate objective for the following analysis.

Under the assumption that prompt $q$ is drawn from a continuous distribution $P(Q)$ and completions $o$ are sampled from the behavioral policy $\pi_{\theta_{\text{old}}}(\cdot|q)$, we derive the standard GRPO gradient as follows:

$$\nabla \mathcal{J}(\theta) = \mathbb{E}_{q \sim P(Q)} \left[ \mathbb{E}_{o \sim \pi_{\theta_{\text{old}}}(\cdot|q)} \left[ \Psi(q, o) \right] \right]$$
$$= \int P(q) \left[ \int \Psi(q, o) \pi_{\theta_{\text{old}}}(o|q) \, do \right] dq, \quad (3)$$

where $\Psi(q, o)$ represents the core gradient component in-

corporating importance sampling, defined as:

$$\Psi(q, o) = \frac{\pi_\theta(o|q)}{\pi_{\theta_{\text{old}}}(o|q)} A(q, o) \nabla_\theta \log \pi_\theta(o|q). \quad (4)$$

Here, $A(q, o)$ denotes the advantage value, and the ratio $\frac{\pi_\theta(o|q)}{\pi_{\theta_{\text{old}}}(o|q)}$ corrects for the distributional shift between the current and behavioral policies. Note that we present the gradient at the sequence level for notational simplicity. The extension to token-level formulation is straightforward and does not affect our theoretical analysis. This formulation makes explicit that GRPO optimizes an expectation over prompts and their corresponding completions sampled from the behavioral policy.

**Unbiasedness at the Completion Level.** Consider a fixed prompt $q$, the original objective is defined over the completions sampled from $\pi_{\theta_{\text{old}}}(\cdot|q)$:

$$\mathbb{E}_{o \sim \pi_{\theta_{\text{old}}}(\cdot|q)} \left[ \Psi(q, o) \right] = \int \Psi(q, o) \pi_{\theta_{\text{old}}}(o|q) \, do. \quad (5)$$

After applying completion-level pruning, the retained samples follow a modified distribution $\tilde{\pi}_{\theta_{\text{old}}}(\cdot|q)$ with density:

$$\tilde{\pi}_{\theta_{\text{old}}}(o|q) = \frac{(1 - \mathcal{P}_t^o(o)) \pi_{\theta_{\text{old}}}(o|q)}{C_o(q)}, \quad (6)$$

where $C_o(q) = \int (1 - \mathcal{P}_t^o(o)) \pi_{\theta_{\text{old}}}(o|q) \, do \in (0, 1)$ is a normalization constant that ensures $\tilde{\pi}_{\theta_{\text{old}}}(\cdot|q)$ remains a valid probability distribution.

To recover the original expectation from samples drawn from $\tilde{\pi}_{\theta_{\text{old}}}(\cdot|q)$, we apply importance sampling with the density ratio:

$$\mathbb{E}_{o \sim \pi_{\theta_{\text{old}}}(\cdot|q)} \left[ \Psi(q, o) \right] = \mathbb{E}_{o \sim \tilde{\pi}_{\theta_{\text{old}}}(\cdot|q)} \left[ \frac{\pi_{\theta_{\text{old}}}(o|q)}{\tilde{\pi}_{\theta_{\text{old}}}(o|q)} \Psi(q, o) \right]. \quad (7)$$

Substituting the expression for $\tilde{\pi}_{\theta_{\text{old}}}(o|q)$, we obtain the importance sampling rescaling factor:

$$\gamma(o, q) = \frac{\pi_{\theta_{\text{old}}}(o|q)}{\tilde{\pi}_{\theta_{\text{old}}}(o|q)} = \frac{C_o(q)}{1 - \mathcal{P}_t^o(o)}, \quad (8)$$

which yields an unbiased gradient estimation:

$$\mathbb{E}_{o \sim \tilde{\pi}_{\theta_{\text{old}}}(\cdot|q)} \left[ \gamma(o, q) \Psi(q, o) \right] = \mathbb{E}_{o \sim \pi_{\theta_{\text{old}}}(\cdot|q)} \left[ \Psi(q, o) \right]. \quad (9)$$

This shows that sampling from the pruned distribution $\tilde{\pi}_{\theta_{\text{old}}}(\cdot|q)$ with rescaling factor $\gamma(o, q)$ yields the same expected gradient as sampling from the original distribution $\pi_{\theta_{\text{old}}}(\cdot|q)$.

**Unbiasedness at the Prompt Level.** Define the completion-level expectation as $G(q) = \mathbb{E}_{o \sim \tilde{\pi}}[\gamma(o, q) \Psi(q, o)]$.

From the completion-level analysis, we have $G(q) = \mathbb{E}_{o \sim \pi_{\theta_{\text{old}}}}[\Psi(q, o)]$. The full objective is $\mathbb{E}_{q \sim P(Q)}[G(q)]$.

After prompt-level pruning, the retained prompts follow a modified distribution $\tilde{P}(Q)$ with density:

$$\tilde{P}(q) = \frac{(1 - \mathcal{P}_t^q(q))P(q)}{C_q}, \qquad (10)$$

where $C_q = \int (1 - \mathcal{P}_t^q(q))P(q)\, dq \in (0, 1)$ is the normalization constant.

Following the same importance sampling principle, we obtain the rescaling factor:

$$\gamma(q) = \frac{P(q)}{\tilde{P}(q)} = \frac{C_q}{1 - \mathcal{P}_t^q(q)}, \qquad (11)$$

which yields an unbiased gradient estimation:

$$\mathbb{E}_{q \sim \tilde{P}(Q)}[\gamma(q)G(q)] = \mathbb{E}_{q \sim P(Q)}[G(q)]. \qquad (12)$$

This shows that sampling from the pruned distribution $\tilde{P}(Q)$ with rescaling factor $\gamma(q)$ yields the same expected gradient as sampling from the original distribution $P(Q)$.

**Unbiasedness of Hierarchical Pruning.** Composing both levels, the hierarchical pruning satisfies:

$$\begin{aligned}
&\mathbb{E}_{q \sim \tilde{P}(Q)}\left[\gamma(q)\mathbb{E}_{o \sim \tilde{\pi}}[\gamma(o, q)\Psi(q, o)]\right] \\
&= \mathbb{E}_{q \sim P(Q)}\left[\mathbb{E}_{o \sim \pi_{\theta_{\text{old}}}}[\Psi(q, o)]\right] \qquad (13) \\
&= \nabla \mathcal{J}(\theta).
\end{aligned}$$

This result shows that DPPO's hierarchical pruning, when combined with appropriate importance rescaling, yields an unbiased gradient estimator of the original GRPO objective.

### 4.2. Instantiating Pruning Policies in Practice

Having established unbiased gradient estimation under continuous distributions, we now instantiate DPPO in the discrete setting in practical training. We first specify concrete pruning criteria for $\mathcal{P}_t^o$ and $\mathcal{P}_t^q$, and then derive the corresponding discrete rescaling factors $\gamma(o, q)$ and $\gamma(q)$.

**Advantage-Aware Completion Pruning.** Lin et al. (2025) observed that a completion's contribution to policy training is related to its advantage magnitude. Based on this, we design a dynamic pruning policy $\mathcal{P}_t^o$ that targets completions with low-absolute advantage as pruning candidates.

For each prompt $q_k$ in the $t$-th epoch, the model generates $G$ completions $\{o_{1,t}^{q_k}, \ldots, o_{G,t}^{q_k}\}$ with corresponding advantages $\{A_{1,t}^{q_k}, \ldots, A_{G,t}^{q_k}\}$. We define a prompt-specific threshold as the mean absolute advantage:

$$\bar{\mathcal{A}}_t^{q_k} = \frac{1}{G} \sum_{i=1}^{G} |A_{i,t}^{q_k}|. \qquad (14)$$

Completions below this threshold are pruned with probability $r_o \in (0, 1)$, while those above are always retained:

$$\mathcal{P}_t^o(o_{i,t}^{q_k}) = \begin{cases} r_o, & |A_{i,t}^{q_k}| \le \bar{\mathcal{A}}_t^{q_k}, \\ 0, & |A_{i,t}^{q_k}| > \bar{\mathcal{A}}_t^{q_k}, \end{cases} \qquad (15)$$

where $r_o \in [0, 1)$ is a predefined hyperparameter. This mechanism ensures that low-value completions still have a chance of being retained.

**History-Guided Prompt Pruning.** Unlike completion-level pruning, which operates after rollout, prompt-level pruning faces a fundamental causality dilemma: assessing a prompt's utility requires generating its completions, which incurs the very cost we aim to reduce. To resolve this, we approximate the current importance using a historical difficulty score $\mathcal{H}_t(q_k)$ based on each prompt's average absolute advantage from the previous epoch.

However, this introduces a new challenge: prompts pruned in the previous epoch receive no new feedback, causing their statistics to become stale. If left unaddressed, such prompts would remain perpetually pruned, leading to biased sampling. We address this with a carry-forward mechanism:

$$\mathcal{H}_t(q_k) = \begin{cases} \bar{\mathcal{A}}_{t-1}^{q_k}, & \text{if } q_k \in \mathcal{S}_{t-1}^q, \\ \mathcal{H}_{t-1}(q_k), & \text{if } q_k \notin \mathcal{S}_{t-1}^q, \end{cases} \qquad (16)$$

where $\mathcal{S}_{t-1}^q$ denotes the set of prompts selected in epoch $t - 1$. If a prompt was selected, its score is updated with fresh statistics; otherwise, we retain its most recent score.

For threshold determination, we adopt a ranking-based strategy. Specifically, we rank prompts by their historical scores $\mathcal{H}_t(q_k)$ in ascending order and select the lowest 50% as pruning candidates:

$$\mathcal{P}_t^q(q_k) = \begin{cases} r_q, & q_k \in \mathcal{B}_t, \\ 0, & q_k \notin \mathcal{B}_t, \end{cases} \qquad (17)$$

where $\mathcal{B}_t$ denotes the set of prompts with the lowest 50% historical scores, and $r_q \in [0, 1)$ is a predefined hyperparameter as the prompt-level pruning probability. Prompt-level pruning is skipped in the first epoch to initialize the historical statistics.

**Discrete Importance Rescaling.** Given the above pruning policies, we now derive the discrete rescaling factors that are applied to retained prompts and completions to ensure unbiased gradient estimation, as established in Section 4.1.

Recall the continuous constants $C_q$ and $C_o(q)$ defined in Equations 6 and 10. In the discrete setting, we have $\mathbb{E}[C_q] = \frac{|\mathcal{S}_t^q|}{|\mathcal{Q}|}$ and $\mathbb{E}[C_o(q_k)] = \frac{|\mathcal{S}_t^o(q_k)|}{|\mathcal{O}|}$, where $|\mathcal{Q}|$, $|\mathcal{O}|$ and $|\mathcal{S}_t^q|$, $|\mathcal{S}_t^o(q_k)|$ denote the number of prompts and completions before and after pruning, respectively. In practice, we set

$C_q = \frac{|\mathcal{S}_t^q|}{|\mathcal{Q}|}$ and $C_o(q_k) = \frac{|\mathcal{S}_t^o(q_k)|}{|\mathcal{O}|}$. Substituting into the theoretical rescaling factors $\gamma(q) = \frac{C_q}{1-\mathcal{P}_t^q(q)}$ and $\gamma(o,q) = \frac{C_o(q)}{1-\mathcal{P}_t^o(o)}$, we obtain:

$$\gamma_t(q_k) = \frac{|\mathcal{S}_t^q|}{|\mathcal{Q}|} \cdot \frac{1}{1 - \mathcal{P}_t^q(q_k)},$$

$$\gamma_t(o_{i,t}^{q_k}, q_k) = \frac{|\mathcal{S}_t^o(q_k)|}{|\mathcal{O}|} \cdot \frac{1}{1 - \mathcal{P}_t^o(o_{i,t}^{q_k})}. \quad (18)$$

When no pruning is applied, the rescaling factors reduce to $\gamma_t(q_k) = \gamma_t(o_{i,t}^{q_k}, q_k) = 1$, which recovers the standard GRPO training. The detailed algorithm is presented in Appendix A.

The final optimization objective of DPPO can be formally expressed as follows:

$$\mathcal{J}_{\text{DPPO}}(\theta) = \mathbb{E}_{q \sim \tilde{P}(Q), \{o_i\}_{i=1}^{|\mathcal{S}_t^o(q)|} \sim \tilde{\pi}_{\theta_{old}}(\cdot|q)} \left[ \gamma_t(q) \right.$$

$$\frac{1}{|\mathcal{S}_t^o(q)|} \sum_{o_i \in \mathcal{S}_t^o(q)} \gamma_t(o_{i,t}^q, q) \left( \frac{1}{|o_i|} \sum_{\tau=1}^{|o_i|} \right.$$

$$\min \left( \rho_{i,\tau}(\theta) A_i, \text{clip}(\rho_{i,\tau}(\theta), 1-\epsilon, 1+\epsilon) A_i \right)$$

$$\left. \left. - \beta \, \mathbb{D}_{KL}(\pi_\theta \| \pi_{ref}) \right) \right]. \quad (19)$$

where $q \sim \tilde{P}(Q)$ is a prompt sampled from the pruned prompt distribution, and $\{o_i\}_{i=1}^{|\mathcal{S}_t^o(q)|}$ are the retained completions drawn from the pruned behavior policy $\tilde{\pi}_{\theta_{\text{old}}}(\cdot|q)$. Here, $\mathcal{S}_t^o(q)$ denotes the retained completion set for prompt $q$ at step $t$. The coefficients $\gamma_t(q)$ and $\gamma_t(o_{i,t}^q, q)$ are the prompt-level and completion-level importance rescaling factors, respectively, which correct the sampling bias introduced by hierarchical pruning.

### 4.3. Dense Prompt Packing for Throughput Preservation

Although our DPPO well filters out redundant computations, it introduces data sparsity. Specifically, filtering out low-value prompts and completions reduces the effective batch size, leading to fragmented memory access and suboptimal GPU occupancy. To counterbalance this side effect and maximize computational throughput, we introduce a Dense Prompt Packing strategy with a window-based greedy selection mechanism, as shown in Figure 4.

Our approach reorganizes the dataset initialization phase to ensure high hardware saturation. Let $L_{max}$ denote the maximum sequence length, defined by the longest sample in the dataset. We maintain a candidate pool $\mathcal{W}$ of size

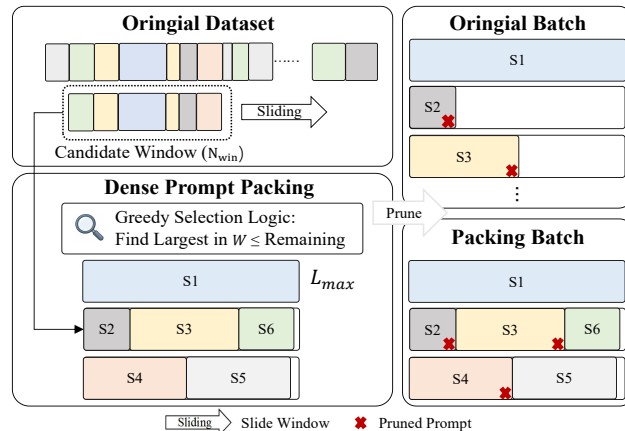

*Figure 4.* Dense Prompt Packing Strategy. The left panel shows the window-based greedy algorithm for assembling variable-length prompts into compact sequences. The right panel indicates the mitigation of pruning-induced sparsity. Unlike standard batching, our approach maximizes valid token density and hardware saturation, ensuring throughput remains consistent with the full-batch pattern.

$N_{win}$, which is continuously replenished from the original unpruned dataset. The packing process operates iteratively: for each packing slot, we initialize a current sequence list and greedily select samples from $\mathcal{W}$ that fit within the remaining space of $L_{max}$. Once a sample is packed, it is removed from $\mathcal{W}$, and a new sample is fetched from the dataset to maintain the pool size. This process repeats until the entire dataset is processed. By aggregating multiple shorter prompts into a single sequence slot, this strategy significantly amplifies the density of valid tokens per batch. This maximization of effective computational load allows the system to maintain high hardware saturation, thereby further accelerating the overall training process.

## 5. Experiments

### 5.1. Experimental Settings

**Models** & **Datasets.** We conduct our main experiments on Qwen3-4B and Qwen3-8B, with additional experiments on Qwen2.5-7B-Instruct, Llama3.2-3B-Instruct, Qwen3-32B, and Qwen3-30B-A3B-Instruct (MoE) presented in the Appendix B. For training data, we use two mathematical reasoning datasets: GSM8K (Cobbe et al., 2021), which comprises 8.5K grade-school problems, and MATH (Hendrycks et al., 2021), which consists of 7.5K competition-level problems.

**Training** & **Evaluation.** We implement DPPO within the verl framework (Sheng et al., 2025), leveraging the vLLM library (Kwon et al., 2023) for efficient rollout generation. All models are trained on 8 NVIDIA H100 GPUs, each equipped with 80GB of SXM. Unless otherwise specified, we adhere to the default hyperparameters from the verl, with the training epochs set to 15, learning rate to $1 \times 10^{-6}$. The policy model temperature is 1, the rollout size is 5, and the

*Table 1.* Comparison between GRPO and DPPO on GSM8K and Math test subset for Qwen3-4B and Qwen3-8B, with DPPO evaluated under varying prompt-level ($r_q$) and completion-level ($r_o$) pruning ratios. Blue (red) values indicate gains (drops) in accuracy relative to the GRPO baseline. Speedup is computed w.r.t. GRPO time (1.00×). GPU-h denotes GPU hours.

| Setting | Method | $r_q$ | $r_o$ | GSM8K Dataset | | | Math Dataset | | |
|---|---|---|---|---|---|---|---|---|---|
| | | | | GPU-h | Speedup | Acc (%) | GPU-h | Speedup | Acc (%) |
| Qwen3-4B | GRPO | 0% | 0% | 64.68 | 1.00× | 93.85 | 137.04 | 1.00× | 77.00 |
| | DPPO | 50% | 50% | 39.28 | 1.65× | $93.49^{-0.36}$ | 88.32 | 1.55× | $78.10^{+1.10}$ |
| | DPPO | 0% | 50% | 49.47 | 1.31× | $94.01^{+0.16}$ | 110.80 | 1.24× | $80.02^{+3.02}$ |
| | DPPO | 50% | 0% | 45.74 | 1.41× | $94.24^{+0.37}$ | 102.48 | 1.34× | $77.65^{+0.65}$ |
| | DPPO | 30% | 30% | 45.94 | 1.41× | $93.86^{+0.01}$ | 101.12 | 1.36× | $77.75^{+0.75}$ |
| | DPPO | 70% | 70% | 32.15 | 2.01× | $94.55^{+0.70}$ | 67.61 | 2.03× | $79.98^{+2.98}$ |
| | DPPO | 90% | 90% | 26.00 | 2.49× | $94.32^{+0.47}$ | 57.88 | 2.37× | $80.15^{+3.15}$ |
| Qwen3-8B | GRPO | 0% | 0% | 78.93 | 1.00× | 94.85 | 137.44 | 1.00× | 77.42 |
| | DPPO | 50% | 50% | 47.00 | 1.68× | $95.15^{+0.30}$ | 94.56 | 1.45× | $79.10^{+1.68}$ |
| | DPPO | 0% | 50% | 56.91 | 1.39× | $94.62^{-0.23}$ | 119.28 | 1.15× | $78.62^{+1.20}$ |
| | DPPO | 50% | 0% | 55.12 | 1.43× | $95.08^{+0.23}$ | 113.20 | 1.21× | $77.99^{+0.57}$ |
| | DPPO | 30% | 30% | 53.73 | 1.47× | $95.08^{+0.23}$ | 119.20 | 1.15× | $79.04^{+1.62}$ |
| | DPPO | 70% | 70% | 39.94 | 1.98× | $95.46^{+0.61}$ | 79.44 | 1.73× | $80.85^{+3.43}$ |
| | DPPO | 90% | 90% | 29.76 | 2.65× | $95.38^{+0.53}$ | 72.32 | 1.90× | $79.12^{+1.70}$ |

*Table 2.* Comparison of DPPO ($r_q = 0.9$, $r_o = 0.9$) and baseline RL methods on different mathematical benchmarks, evaluated on Qwen3-4B and Qwen3-8B in terms of accuracy and speedup (GRPO=1.00×). **Bold** indicates the best result in each column.

| Model | Method | Speedup ↑ | MATH500 | AIME25 | AIME24 | AMC | Minerva. | Olympiad. | Avg. ↑ |
|---|---|---|---|---|---|---|---|---|---|
| Qwen3-4B | GRPO | 1.00× | 76.20 | 8.33 | 7.50 | 45.78 | 29.41 | 34.72 | 33.44 |
| | CPPO | 1.35× | 75.15 | 6.67 | 5.83 | 41.87 | 28.49 | 31.49 | 31.58 |
| | GRESO | 2.31× | 76.40 | 10.83 | 10.83 | **47.29** | **30.15** | 37.28 | 35.10 |
| | DPPO | **2.37×** | **77.90** | **11.67** | **17.50** | 46.99 | 29.96 | **38.02** | **36.80** |
| Qwen3-8B | GRPO | 1.00× | 75.80 | 8.33 | 7.50 | 46.08 | 29.32 | 34.27 | 33.55 |
| | CPPO | 1.32× | 75.40 | 6.67 | 5.83 | 41.57 | 29.23 | 31.53 | 31.71 |
| | GRESO | 1.88× | 76.65 | 10.83 | 10.83 | **47.29** | **29.78** | 37.35 | 35.45 |
| | DPPO | **1.90×** | **77.10** | **11.67** | **17.50** | 46.69 | 29.04 | **38.06** | **36.87** |

maximum completion length is 1024. We use Pass@1 accuracy as the evaluation metric. To evaluate the generalization, we train models on the MATH dataset and evaluate them on other mathematical reasoning benchmarks, including Math500 (Hendrycks et al., 2021; Lightman et al., 2024), AIME2025 (Mathematical Association of America, 2025), AIME2024 (Mathematical Association of America, 2024), AMC 2023 (Mathematical Association of America, 2023), Minerva Math (Lewkowycz et al., 2022), and Olympiad Bench (He et al., 2024).

### 5.2. Main Results

**GSM8K & Math.** As shown in Table 1, DPPO consistently accelerates training while maintaining or even improving accuracy across both datasets. On GSM8K, DPPO achieves up to 2.65× speedup on Qwen3-8B with a +0.53% accuracy gain, and up to 2.49× speedup on Qwen3-4B while preserving baseline performance. On the more challenging MATH dataset, the efficiency-accuracy trade-off is even more favorable: for Qwen3-4B, the most aggressive pruning configuration ($r_q = 0.9, r_o = 0.9$) yields a 2.37× speedup alongside a +3.15% accuracy improvement. For Qwen3-8B, DPPO achieves up to 1.90× speedup with a +1.70% accuracy gain. These results show that DPPO eliminates computational redundancy without compromising reasoning capabilities, and in many cases, the selective retention of high-value samples leads to improved model performance.

**Out-of-Distribution Evaluation.** We compare DPPO

against GRPO (Guo et al., 2025) and several heuristic pruning baselines: GRESO (Zheng et al., 2025b) and CPPO (Lin et al., 2025). As shown in Table 2, DPPO consistently outperforms all baselines across both model scales. On Qwen3-4B, DPPO achieves an average accuracy of 36.80%, surpassing GRPO by +3.36%, CPPO by +5.22%, and GRESO by +1.70%, while achieving a 2.37× speedup. The improvements are particularly pronounced on competition-level benchmarks such as AIME24 (+10.00% over GRPO) and Olympiad (+3.30% over GRPO), demonstrating the effectiveness of our importance sampling-based pruning strategy in enhancing complex reasoning capabilities. On the larger Qwen3-8B model, DPPO further amplifies these gains, achieving an average accuracy of 36.87% with a 1.90× speedup over GRPO. Notably, DPPO outperforms all baselines on 4 out of 6 benchmarks, with the largest improvement observed on AIME24 (+10.00% over GRPO). These results confirm that DPPO not only accelerates training but also improves generalization to out-of-distribution mathematical reasoning tasks. We also provide a case study with a concrete mathematical example in Appendix E.

**Robustness Across RL Algorithms.** DPPO is designed as a general-purpose framework that can be combined with diverse RL training pipelines. To validate its broad applicability beyond GRPO, we evaluate DPPO in combination with DAPO (Yu et al., 2025) and GSPO (Zheng et al., 2025a). DAPO is a recent RL algorithm with decoupled clipping and dynamic sampling, while GSPO is a sequence-level optimization method based on likelihood ratios. In Table 3, integrating DPPO into DAPO and GSPO yields consistent efficiency gains: On Qwen3-4B, DPPO achieves 1.98× speedup over DAPO while maintaining accuracy (+0.01%); on Qwen3-8B, DPPO accelerates GSPO by 2.46× with marginal accuracy difference (-0.15%). These results confirm that our importance-sampling-based pruning strategy is algorithm-agnostic. Additional experiments on DAPO, GSPO, as well as larger-scale models and different model families are provided in Appendix B.2, where DPPO achieves up to 4.87× speedup without accuracy degradation.

### 5.3. Ablation Study

**Ablation of Rollout Size.** To investigate the impact of rollouts, we conduct experiments on GSM8K with Qwen3-4B under varying rollout sizes $G \in \{8, 16\}$, as presented in Table 4. We observe that: (1) DPPO consistently accelerates training across different rollout settings, achieving 1.64× speedup at $G = 8$ and 1.61× speedup at $G = 16$; (2) the accuracy gap between GRPO and DPPO remains minimal (within 0.37%) regardless of rollout count, confirming that our importance-sampling-based correction effectively preserves gradient quality; (3) larger rollout counts provide more candidates for pruning, potentially enabling more aggressive sample selection while maintaining performance.

*Table 3.* Generalization of DPPO when integrated with DAPO on GSM8K for Qwen3-4B and Qwen3-8B, with prompt-level and completion-level pruning ratios set to $r_q = 0.9$ and $r_o = 0.9$.

| Setting | Method | GPU-h | Speedup | Acc (%) |
|---------|--------|-------|---------|---------|
| Qwen3-4B | DAPO | 48.53 | 1.00× | 94.09 |
| | + DPPO | 24.57 | 1.98× | 94.10 |
| | GSPO | 61.40 | 1.00× | 93.78 |
| | + DPPO | 25.07 | 2.45× | 94.17 |
| Qwen3-8B | DAPO | 55.24 | 1.00× | 95.14 |
| | + DPPO | 28.72 | 1.92× | 94.85 |
| | GSPO | 66.19 | 1.00× | 94.85 |
| | + DPPO | 26.88 | 2.46× | 94.70 |

*Table 4.* Ablation study on the impact of rollout numbers ($G$). We report the accuracy and computational cost of Qwen3-4B on GSM8K under different group sizes with prompt-level and completion-level pruning ratios set to $r_q = 0.5$ and $r_o = 0.5$.

| Method | Rollout ($G$) | GPU-h | Speedup | Acc (%) |
|--------|---------------|-------|---------|---------|
| GRPO | 8 | 89.76 | 1.00× | 94.84 |
| DPPO | 8 | 54.80 | 1.64× | 94.47 |
| GRPO | 16 | 173.36 | 1.00× | 94.69 |
| DPPO | 16 | 107.76 | 1.61× | 94.47 |

These results demonstrate that DPPO is robust to the choice of rollout count and can be flexibly combined with different sampling configurations.

**Ablation of Pruning Rules.** To validate our pruning criterion, we compare it against a reverse strategy that prunes in the opposite direction and a random strategy that prunes samples without any criterion. All variants share the same rescaling, so that any performance difference reflects the selection rule alone rather than the rescaling operation. As shown in Table 5, the random strategy matches the baseline almost exactly (93.82% vs. 93.85% at the completion level, and 93.90% vs. 93.85% at the prompt level), indicating that rescaling preserves the unbiasedness of the estimator and that indiscriminate pruning by itself neither helps nor harms. The gain of our method therefore cannot be attributed to rescaling alone. Building on this unbiased foundation, our criterion improves accuracy to 94.01% (+0.16%) for completion-level pruning and 94.24% (+0.39%) for prompt-level pruning, whereas the reverse strategy degrades it to 93.52% and 93.08%, respectively. These results disentangle the two sources of benefit: rescaling maintains unbiasedness (Random ≈ Baseline), while our criterion contributes the actual gain by retaining high-advantage completions and prompts, thereby inducing an implicit curriculum.

**Ablation Study on the Key Modules.** As shown in Table 6, by discarding low-informative completions, the completion-

*Table 5.* Ablation study on pruning criteria, evaluated on GSM8K with Qwen3-4B. 'Reverse' retains the samples that our method would prune.

| Module | Pruning Rules | $r_q$ | $r_o$ | Acc (%) |
|---|---|---|---|---|
| **Baseline (No Pruning)** | – | – | – | 93.85 |
| *Completion-level Ablation* | | | | |
| **Ours (Prune Low)** | $\|A_{i,t}^{q_k}\| \leq \bar{\mathcal{A}}_t^{q_k}$ | 0 | 0.5 | 94.01 |
| Reverse (Prune High) | $\|A_{i,t}^{q_k}\| > \bar{\mathcal{A}}_t^{q_k}$ | 0 | 0.5 | 93.52 |
| Random | Random | - | - | 93.82 |
| *Prompt-level Ablation* | | | | |
| **Ours (Prune Low)** | $q_k \in \mathcal{B}_t$ | 0.5 | 0 | 94.24 |
| Reverse (Prune High) | $q_k \notin \mathcal{B}_t$ | 0.5 | 0 | 93.08 |
| Random | Random | - | - | 93.90 |

*Table 6.* Ablation study on the incremental contribution of each component ($r_q = 0.5$, $r_o = 0.5$), evaluated on GSM8K with Qwen3-8B. 'Comp.' and 'Prompt' denote Completion-level and Prompt-level pruning, respectively.

| Method | Training Time (GPU hours) | Speedup | Acc (%) |
|---|---|---|---|
| Baseline | 78.93 | 1.00× | 94.85 |
| + Comp. Pruning | 55.10 | 1.34× | 94.77 |
| + Prompt Pruning | 48.79 | 1.52× | 94.94 |
| + Dense Packing | 47.00 | 1.68× | 95.15 |

level pruning module improves training efficiency by 1.34× with only -0.08% accuracy drop. By further filtering easy prompts, prompt-level pruning increases the speedup to 1.52×. Finally, by fully leveraging GPU parallel computing capability through dense packing, the overall training efficiency reaches 1.68× with comparable performance, demonstrating the complementary benefits of all three components.

## 6. Conclusion

In this work, we presented Dynamic Pruning Policy Optimization (DPPO), a theoretically rigorous framework to accelerate GRPO without compromising gradient estimation accuracy. By reformulating selective data utilization as a hierarchical importance sampling process, DPPO eliminates redundancy at both prompt and completion levels while correcting for distributional shifts to maintain an unbiased optimization objective. Also, we addressed the practical challenge of pruning-induced sparsity through a Dense Prompt Packing strategy, which ensures high valid token density and optimal hardware saturation. Extensive empirical evaluations show that DPPO well reduces computational overhead while preserving model performance, establishing a new paradigm for scalable, efficient, and mathematically grounded reinforcement learning in large language models.

## Acknowledgements

This work was supported by National Natural Science Foundation of China 623B2016 and 62576018, Zhejiang Provincial Natural Science Foundation of China under Grant No. LD24F020007, Beijing Natural Science Foundation L244043, and Taiyuan City "Double hundred Research action" 2024TYJB0127.

## Impact Statement

This paper presents work whose goal is to advance the field of machine learning, specifically improving the computational efficiency of reinforcement learning for large language models. There are many potential societal consequences of our work, none of which we feel must be specifically highlighted here.

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

# Appendix

## A. Algorithm

We provide the algorithm for our Dynamic Pruning Policy Optimization (DPPO) in Algorithm 1. During dataset initialization, we apply Dense Prompt Packing to aggregate multiple prompts into compact sequences, maximizing GPU utilization throughout training. This effectively increases the number of prompts per batch without additional memory overhead, equivalent to enlarging the batch size in a length-aware manner. For prompt-level pruning, we perform filtering at the beginning of each batch based on historical statistics $\mathcal{H}_t(q_k)$ collected from the previous epoch, which avoids the causality dilemma where assessing a prompt's value would require generating completions first. Since no historical information is available initially, prompt-level pruning is skipped during the first epoch. For completion-level pruning, we compute a per-prompt threshold $\bar{\mathcal{A}}_t^{q_k}$ as the mean absolute advantage, and deterministically prune a fraction $r_o$ of completions with $|A_{i,t}^{q_k}| \leq \bar{\mathcal{A}}_t^{q_k}$. While $r_q$ and $r_o$ denote per-sample pruning probabilities, we implement this deterministically by pruning the corresponding fraction of each candidate set, reducing variance while preserving the expected pruning rate. The retained samples are then reweighted using importance sampling factors $\gamma_t(q_k)$ and $\gamma_t(o_{i,t}^{q_k}, q_k)$ to ensure unbiased gradient estimation.

---

**Algorithm 1** Dynamic Pruning Policy Optimization (DPPO)

---

**Require:** Initial model $\pi_{\theta_{\text{init}}}$; dataset $\mathcal{D}$ with prompts $\mathcal{Q}$; group size $G$; pruning rates $r_o, r_q \in [0, 1)$; max sequence length $L_{\max}$; window size $N_{\text{win}}$
  1: Initialize $\pi_\theta \leftarrow \pi_{\theta_{\text{init}}}$, $\pi_{\text{ref}} \leftarrow \pi_\theta$, $\mathcal{H}_0(q_k) \leftarrow 0$ for all $q_k \in \mathcal{Q}$
  2: # Dense Prompt Packing (Dataset Initialization)
  3: Pack prompts in $\mathcal{D}$ into sequences of length $\leq L_{\max}$ via window-based greedy selection with candidate pool $\mathcal{W}$ of size $N_{\text{win}}$
  4: **for** epoch $t = 1, \ldots, T$ **do**
  5:     $\pi_{\theta_{\text{old}}} \leftarrow \pi_\theta$
  6:     **for** each batch $\mathcal{P}$ from $\mathcal{D}$ **do**
  7:       # Prompt-Level Pruning (skip if $t = 1$)
  8:       Let $\mathcal{B}_t \leftarrow$ bottom 50% of prompts in $\mathcal{P}$ ranked by $\mathcal{H}_{t-1}(q_k)$
  9:       Prune $\lfloor r_q \cdot |\mathcal{B}_t| \rfloor$ prompts from $\mathcal{B}_t$; let $\mathcal{S}_t^q$ be all retained prompts in $\mathcal{P}$ (Eq. 17)
10:       # Rollout
11:       **for** each retained prompt $q_k \in \mathcal{S}_t^q$ **do**
12:         Sample $G$ completions $\{o_{1,t}^{q_k}, \ldots, o_{G,t}^{q_k}\}$ from $\pi_{\theta_{\text{old}}}(\cdot|q_k)$
13:         Compute advantages $\{A_{i,t}^{q_k}\}_{i=1}^G$
14:       **end for**
15:       # Completion-Level Pruning
16:       **for** each prompt $q_k \in \mathcal{S}_t^q$ **do**
17:         Compute threshold $\bar{\mathcal{A}}_t^{q_k} \leftarrow \frac{1}{G} \sum_{i=1}^G |A_{i,t}^{q_k}|$ (Eq. 14)
18:         Let $\mathcal{C}_t^{q_k} = \{o_{i,t}^{q_k} : |A_{i,t}^{q_k}| \leq \bar{\mathcal{A}}_t^{q_k}\}$ be the pruning candidates
19:         Prune $\lfloor r_o \cdot |\mathcal{C}_t^{q_k}| \rfloor$ completions from $\mathcal{C}_t^{q_k}$; let $\mathcal{S}_t^o(q_k)$ be the retained completions (Eq. 15)
20:         Update historical score $\mathcal{H}_t(q_k) \leftarrow \bar{\mathcal{A}}_t^{q_k}$ (Eq. 16)
21:       **end for**
22:       # Policy Update with Importance Rescaling
23:       Update $\pi_\theta$ using reweighted gradients with factors $\gamma_t(q_k) \cdot \gamma_t(o_{i,t}^{q_k}, q_k)$ (Eq. 18)
24:     **end for**
25: **end for**
**Ensure:** $\pi_\theta$

---

# B. Experiments

## B.1. Experiments settings

To evaluate generalization on out-of-distribution datasets, we adopt the following evaluation protocol: for each prompt, we sample 4 completions with temperature 0.7 and maximum response length of 1024, consistent with training. We use Pass@1

*Table 7.* Comparison between DAPO and DPPO on GSM8K for Qwen3-4B and Qwen3-8B, with DPPO evaluated under varying prompt-level ($r_q$) and completion-level ($r_o$) pruning ratios. Accuracy changes relative to DAPO baseline are shown in superscript (blue for improvement, red for decline).

| Setting | Method | $r_q$ | $r_o$ | GPU-h | Speedup | Acc (%) |
|---|---|---|---|---|---|---|
| Qwen3-4B | DAPO | 0% | 0% | 48.53 | 1.00× | 94.09 |
| | + DPPO | 50% | 50% | 32.43 | 1.50× | $94.02^{-0.07}$ |
| | + DPPO | 0% | 50% | 42.90 | 1.13× | $94.24^{+0.15}$ |
| | + DPPO | 50% | 0% | 42.45 | 1.14× | $94.24^{+0.15}$ |
| | + DPPO | 30% | 30% | 37.27 | 1.30× | $94.70^{+0.61}$ |
| | + DPPO | 70% | 70% | 31.74 | 1.53× | $94.70^{+0.61}$ |
| | + DPPO | 90% | 90% | 24.57 | 1.98× | $94.10^{+0.01}$ |
| Qwen3-8B | DAPO | 0% | 0% | 55.24 | 1.00× | 95.14 |
| | + DPPO | 50% | 50% | 40.15 | 1.38× | $95.38^{+0.24}$ |
| | + DPPO | 0% | 50% | 44.70 | 1.24× | $95.15^{+0.01}$ |
| | + DPPO | 50% | 0% | 44.14 | 1.25× | $95.15^{+0.01}$ |
| | + DPPO | 30% | 30% | 44.34 | 1.25× | $95.23^{+0.09}$ |
| | + DPPO | 70% | 70% | 32.93 | 1.68× | $94.55^{-0.59}$ |
| | + DPPO | 90% | 90% | 28.72 | 1.92× | $94.85^{-0.29}$ |

accuracy as the evaluation metric.

## B.2. More Experiments

**Comparison on DAPO.** As shown in Table 7, augmenting DAPO with DPPO consistently accelerates GSM8K training with competitive accuracy. For Qwen3-4B, DPPO reaches 1.98× speedup at $r_q = r_o = 0.9$ while preserving performance (+0.01%), and improves accuracy by up to +0.61% under moderate pruning ($r_q = r_o = 0.3$ or $0.7$) with 1.30×–1.53× speedup. For Qwen3-8B, DPPO achieves up to +0.24% accuracy improvement at $r_q = r_o = 0.5$ with 1.38× speedup, and reaches 1.92× speedup at $r_q = r_o = 0.9$ with only a marginal accuracy drop (-0.29%). Overall, DPPO serves as a plug-and-play efficiency booster for DAPO, substantially reducing GPU-hours with a favorable efficiency–accuracy trade-off.

**Comparison on GSPO.** Table 8 shows that DPPO can be seamlessly integrated into GSPO to substantially reduce training cost on GSM8K while keeping accuracy stable. For Qwen3-4B, DPPO provides consistent speedups ranging from 1.30× to 2.45×, and the best-performing configuration improves accuracy by +0.39%. For Qwen3-8B, the gains are more pronounced: DPPO reaches up to 2.46× speedup, and moderate-to-high pruning yields simultaneous improvements in efficiency and accuracy, peaking at +0.37% with a 2.21× speedup. Even under the most aggressive setting, the accuracy change remains small (-0.15%), indicating strong robustness of the pruning strategy. Overall, DPPO strengthens GSPO with a favorable cost–performance profile, suggesting that selectively focusing updates on informative samples can remove redundant computation without hurting downstream reasoning quality.

**Comparison across Different Models.** Table 9 verifies that DPPO is broadly compatible with different RL optimizers on MATH using Qwen2.5-7B-Instruct and Llama3.2-3B-Instruct. When integrated into GRPO, DPPO provides consistent acceleration up to 2.08× while keeping accuracy stable, and even improves performance by as much as +0.63%. Notably, strong pruning remains effective: at $r_q = r_o = 0.9$, DPPO attains a 2.08× speedup together with a +0.49% gain, indicating that a large portion of computation can be removed without hurting reasoning quality. When plugged into DAPO, DPPO achieves 1.14×–1.63× speedup and yields accuracy improvements across all settings, reaching up to +1.09%. Overall, these results demonstrate the versatility of DPPO as a plug-and-play pruning strategy that consistently reduces training cost while preserving or improving performance across different models and RL algorithms.

**Scaling to Larger Models.** Table 10 demonstrates that DPPO scales effectively to larger models, delivering consistent efficiency gains alongside notable accuracy improvements on the MATH dataset. For Qwen3-32B, DPPO achieves speedups ranging from 1.23× to 2.57× while improving accuracy across all pruning configurations, with gains from +0.37% to +4.19%. Notably, aggressive pruning does not degrade performance: at $r_q = r_o = 0.7$, DPPO attains a 2.02× speedup with the largest accuracy gain of +4.19%, and even at $r_q = r_o = 0.9$, it maintains a +3.30% improvement with 2.57×

*Table 8.* Comparison between GSPO and DPPO on GSM8K for Qwen3-4B and Qwen3-8B, with DPPO evaluated under varying prompt-level ($r_q$) and completion-level ($r_o$) pruning ratios. Accuracy changes relative to GSPO baseline are shown in superscript (blue for improvement, red for decline).

| Setting | Method | $r_q$ | $r_o$ | GPU-h | Speedup | Acc (%) |
|---------|--------|-------|-------|-------|---------|---------|
| Qwen3-4B | GSPO | 0% | 0% | 61.40 | $1.00\times$ | 93.78 |
| | + DPPO | 50% | 50% | 35.85 | $1.71\times$ | $93.49^{-0.29}$ |
| | + DPPO | 0% | 50% | 47.09 | $1.30\times$ | $94.17^{+0.39}$ |
| | + DPPO | 50% | 0% | 44.64 | $1.38\times$ | $94.17^{+0.39}$ |
| | + DPPO | 30% | 30% | 42.85 | $1.43\times$ | $94.17^{+0.39}$ |
| | + DPPO | 70% | 70% | 27.84 | $2.21\times$ | $93.94^{+0.16}$ |
| | + DPPO | 90% | 90% | 25.06 | $2.45\times$ | $94.17^{+0.39}$ |
| Qwen3-8B | GSPO | 0% | 0% | 66.19 | $1.00\times$ | 94.85 |
| | + DPPO | 50% | 50% | 36.65 | $1.81\times$ | $94.85^{+0.00}$ |
| | + DPPO | 0% | 50% | 46.01 | $1.44\times$ | $95.00^{+0.15}$ |
| | + DPPO | 50% | 0% | 49.36 | $1.34\times$ | $94.85^{+0.00}$ |
| | + DPPO | 30% | 30% | 45.37 | $1.46\times$ | $95.08^{+0.23}$ |
| | + DPPO | 70% | 70% | 29.98 | $2.21\times$ | $95.22^{+0.37}$ |
| | + DPPO | 90% | 90% | 26.88 | $2.46\times$ | $94.70^{-0.15}$ |

*Table 9.* Robustness across RL algorithms. We integrate DPPO into GRPO and DAPO on the MATH dataset using Qwen2.5-7B-Instruct and Llama3.2-3B-Instruct . Accuracy changes relative to each baseline are shown in superscript (blue for improvement, red for decline).

| Setting | Method | $r_q$ | $r_o$ | GPU-h | Speedup | Acc (%) |
|---------|--------|-------|-------|-------|---------|---------|
| Qwen2.5-7B-Instruct | GRPO | 0% | 0% | 58.39 | $1.00\times$ | 71.84 |
| | + DPPO | 50% | 50% | 37.31 | $1.56\times$ | $71.97^{+0.13}$ |
| | + DPPO | 0% | 50% | 41.31 | $1.41\times$ | $72.47^{+0.63}$ |
| | + DPPO | 50% | 0% | 42.84 | $1.36\times$ | $71.76^{-0.08}$ |
| | + DPPO | 30% | 30% | 42.60 | $1.37\times$ | $72.07^{+0.23}$ |
| | + DPPO | 70% | 70% | 32.35 | $1.80\times$ | $71.56^{-0.28}$ |
| | + DPPO | 90% | 90% | 28.11 | $2.08\times$ | $72.33^{+0.49}$ |
| Llama3.2-3B-Instruct | GRPO | 0% | 0% | 41.86 | $1.00\times$ | 49.38 |
| | + DPPO | 50% | 50% | 30.02 | $1.40\times$ | $49.14^{-0.24}$ |
| | + DPPO | 0% | 50% | 33.30 | $1.26\times$ | $49.72^{+0.34}$ |
| | + DPPO | 50% | 0% | 37.30 | $1.12\times$ | $49.38^{+0.00}$ |
| | + DPPO | 30% | 30% | 33.29 | $1.26\times$ | $50.22^{+0.84}$ |
| | + DPPO | 70% | 70% | 27.73 | $1.53\times$ | $49.38^{+0.00}$ |
| | + DPPO | 90% | 90% | 27.21 | $1.54\times$ | $48.96^{-0.42}$ |
| Qwen2.5-7B-Instruct | DAPO | 0% | 0% | 45.95 | $1.00\times$ | 71.18 |
| | + DPPO | 50% | 50% | 35.64 | $1.29\times$ | $71.89^{+0.71}$ |
| | + DPPO | 0% | 50% | 40.47 | $1.14\times$ | $72.27^{+1.09}$ |
| | + DPPO | 50% | 0% | 40.07 | $1.15\times$ | $71.87^{+0.69}$ |
| | + DPPO | 30% | 30% | 38.99 | $1.18\times$ | $72.21^{+1.03}$ |
| | + DPPO | 70% | 70% | 30.89 | $1.49\times$ | $71.64^{+0.46}$ |
| | + DPPO | 90% | 90% | 28.25 | $1.63\times$ | $71.40^{+0.22}$ |

speedup. Similar trends are observed for Qwen3-30B-A3B-Instruct, a Mixture-of-Experts model, where DPPO yields even larger speedups due to the inherent sparsity of MoE architectures. At $r_q = r_o = 0.7$, DPPO achieves a $3.45\times$ speedup while reaching 85.56% accuracy (+3.39%), and at $r_q = r_o = 0.9$, it attains a $4.87\times$ speedup with +3.35% accuracy gain. These consistent results across both dense and MoE architectures suggest that larger models exhibit greater computational redundancy during RL training, which DPPO effectively exploits through selective sample retention.

*Table 10.* Robustness across larger model on Math dataset. Accuracy changes relative to each baseline are shown in superscript (blue for improvement, red for decline).

| Setting | Method | $r_q$ | $r_o$ | GPU-h | Speedup | Acc (%) |
|---------|--------|------|------|-------|---------|---------|
| | GRPO | 0% | 0% | 204.08 | 1.00× | 80.43 |
| | + DPPO | 50% | 50% | 127.92 | 1.60× | $82.55^{+2.12}$ |
| | + DPPO | 0% | 50% | 166.08 | 1.23× | $81.95^{+1.52}$ |
| Qwen3-32B | + DPPO | 50% | 0% | 159.84 | 1.28× | $80.80^{+0.37}$ |
| | + DPPO | 30% | 30% | 157.28 | 1.30× | $81.25^{+0.82}$ |
| | + DPPO | 70% | 70% | 101.20 | 2.02× | $84.62^{+4.19}$ |
| | + DPPO | 90% | 90% | 79.31 | 2.57× | $83.73^{+3.30}$ |
| | GRPO | 0% | 0% | 413.76 | 1.00× | 82.17 |
| | + DPPO | 50% | 50% | 179.52 | 2.30× | $84.94^{+2.77}$ |
| | + DPPO | 0% | 50% | 225.84 | 1.83× | $84.71^{+2.54}$ |
| Qwen3-30B -A3B-Instruct | + DPPO | 50% | 0% | 247.20 | 1.67× | $83.73^{+1.56}$ |
| | + DPPO | 30% | 30% | 234.72 | 1.76× | $84.60^{+2.43}$ |
| | + DPPO | 70% | 70% | 120.07 | 3.45× | $85.56^{+3.39}$ |
| | + DPPO | 90% | 90% | 84.96 | 4.87× | $85.52^{+3.35}$ |

## C. Stability and Convergence

We plot the policy gradient (PG) loss, KL divergence, and reward curves in Figure 5 for Qwen3-4B and Qwen3-8B trained on the MATH dataset.

**PG Loss.** DPPO exhibits fluctuation patterns comparable to GRPO across most pruning configurations, suggesting that hierarchical pruning with importance rescaling preserves stable optimization dynamics. At higher pruning rates ($r_q = 0.9, r_o = 0.9$), slightly increased variance is observed, which aligns with our theoretical expectation: aggressive pruning yields larger rescaling factors, amplifying gradient variance while maintaining unbiased estimation. We provide a detailed variance analysis in Section D.

**Reward.** We observe that higher pruning ratios lead to lower mean training reward. This can be explained by our pruning strategy: samples with low absolute advantage are pruning candidates, and these are precisely the samples where all completions receive similar rewards (either consistently high or consistently low). By pruning them, we retain samples where the model exhibits high uncertainty completions with diverse reward outcomes. These retained samples naturally have lower mean reward but provide richer contrastive signals for policy learning. The result is a model that achieves lower training reward but better generalization, as it focuses on learning from the most informative samples rather than reinforcing already-mastered patterns.

**KL Divergence.** The KL divergence curves reveal that DPPO maintains substantially lower KL divergence compared to GRPO throughout training. This indicates that our pruning strategy, by focusing on high-uncertainty samples, enables more efficient policy updates while staying closer to the reference policy. Combined with the reward analysis above, this further explains DPPO's strong generalization capability, as corroborated by our out-of-distribution evaluation results in Table 2.

## D. Variance Analysis

We analyze the variance of DPPO's gradient estimator under hierarchical pruning.

### D.1. Completion-Level Variance

For a **fixed** prompt $q$, let $G_o = \Psi(q, o)$ and $G(q) = \frac{1}{|\mathcal{O}|} \sum_{o \in \mathcal{O}} G_o$. Recall from Section 4.1 that the rescaling factor is $\gamma(o, q) = \frac{C_o(q)}{1 - \mathcal{P}_t^o(o)}$.

The covariance of the rescaled gradient is:

$$\text{Cov}[\gamma G_o] = \mathbb{E}_{o \sim \tilde{\pi}}[(\gamma G_o)(\gamma G_o)^\top] - \mathbb{E}_{o \sim \tilde{\pi}}[\gamma G_o]\mathbb{E}_{o \sim \tilde{\pi}}[\gamma G_o]^\top = \frac{C_o(q)}{|\mathcal{O}|} \sum_{o \in \mathcal{O}} \frac{G_o G_o^\top}{1 - \mathcal{P}_t^o(o)} - G(q)G(q)^\top. \tag{20}$$

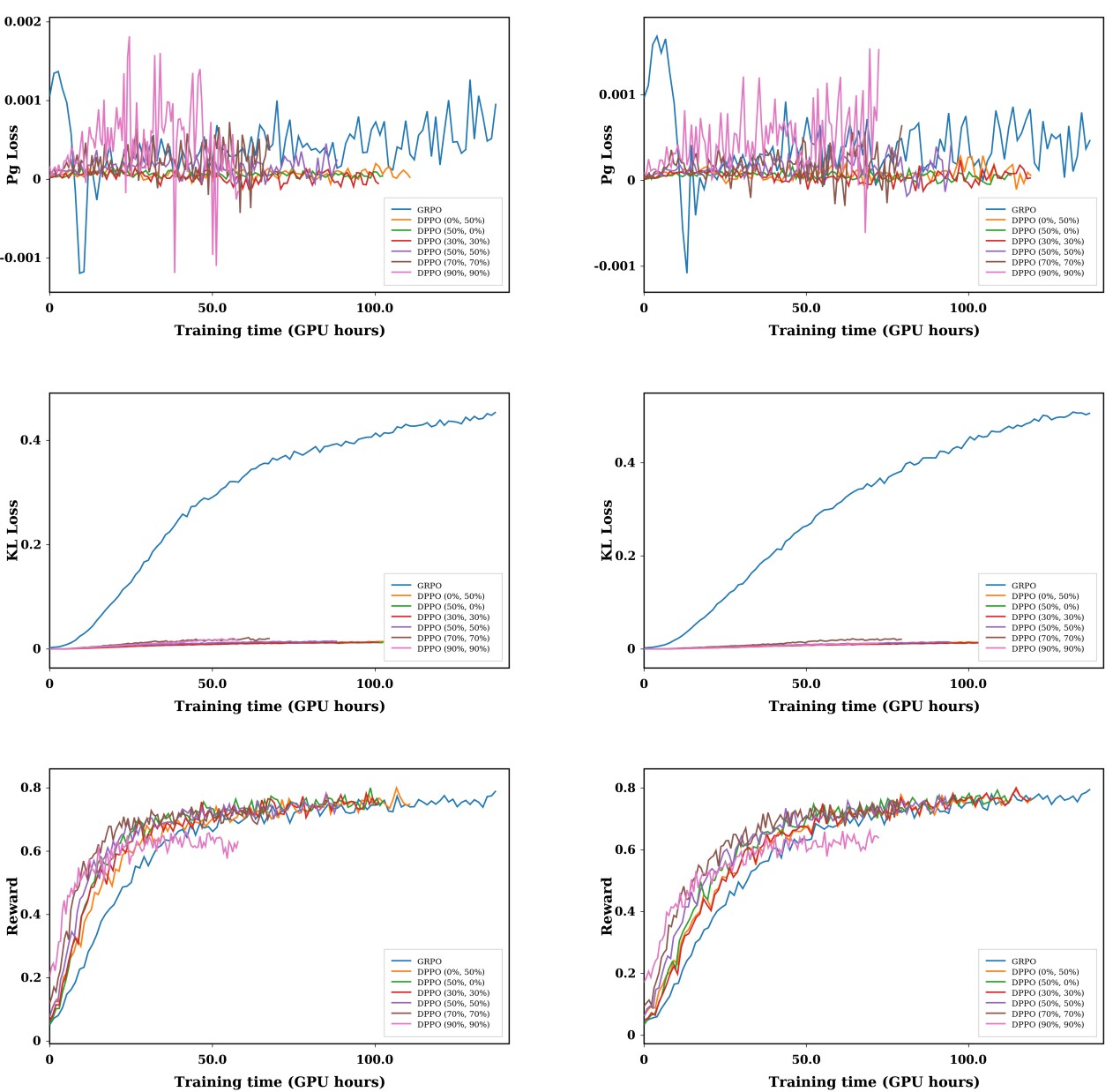

*Figure 5.* Training dynamics of GRPO and DPPO variants on MATH dataset with Qwen3-4B (left) and Qwen3-8B (right).

The diagonal terms give the variance:

$$\text{Var}[\gamma G_o] = \frac{C_o(q)}{|\mathcal{O}|} \sum_{o \in \mathcal{O}} \frac{G_o^2}{1 - \mathcal{P}_t^o(o)} - G(q)^2 = \frac{|\mathcal{S}_t^o|}{|\mathcal{O}|^2} \sum_{o \in \mathcal{O}} \frac{G_o^2}{1 - \mathcal{P}_t^o(o)} - G(q)^2. \tag{21}$$

Using the identity $\frac{1}{1-x} = 1 + \frac{x}{1-x}$:

$$\sum_{o \in \mathcal{O}} \frac{G_o^2}{1 - \mathcal{P}_t^o(o)} = |\mathcal{O}|\mathbb{E}_{\mathcal{O}}[G_o^2] + \sum_{o \in \mathcal{O}} \frac{\mathcal{P}_t^o(o)G_o^2}{1 - \mathcal{P}_t^o(o)}. \tag{22}$$

In DPPO's score-based pruning, $\mathcal{P}_t^o(o) = 0$ for high-score completions (large $|A(q,o)|$), while low-score completions are pruned with a uniform probability $r_o$ (cf. Eq. (15)). Under the condition that low-score samples have smaller gradient, we

have:

$$\mathbb{E}_{o|\mathcal{P}_t^o(o)\neq 0}\left[\frac{G_o^2}{1-\mathcal{P}_t^o(o)}\right] \leq \mathbb{E}_{\mathcal{O}}[G_o^2], \tag{23}$$

then the second term in (22) can be bounded as:

$$\sum_{o\in\mathcal{O}}\frac{\mathcal{P}_t^o(o)G_o^2}{1-\mathcal{P}_t^o(o)} \leq \sum_{o\in\mathcal{O}}\mathcal{P}_t^o(o)\cdot\mathbb{E}_{\mathcal{O}}[G_o^2] = (|\mathcal{O}|-|\mathcal{S}_t^o|)\mathbb{E}_{\mathcal{O}}[G_o^2]. \tag{24}$$

Substituting (24) into (22):

$$\sum_{o\in\mathcal{O}}\frac{G_o^2}{1-\mathcal{P}_t^o(o)} \leq |\mathcal{O}|\mathbb{E}_{\mathcal{O}}[G_o^2] + (|\mathcal{O}|-|\mathcal{S}_t^o|)\mathbb{E}_{\mathcal{O}}[G_o^2] = (2|\mathcal{O}|-|\mathcal{S}_t^o|)\mathbb{E}_{\mathcal{O}}[G_o^2]. \tag{25}$$

Substituting into (21):

$$\mathrm{Var}[\gamma G_o] \leq \frac{|\mathcal{S}_t^o|}{|\mathcal{O}|^2}(2|\mathcal{O}|-|\mathcal{S}_t^o|)\mathbb{E}_{\mathcal{O}}[G_o^2] - G(q)^2 = \frac{|\mathcal{S}_t^o|(2|\mathcal{O}|-|\mathcal{S}_t^o|)}{|\mathcal{O}|^2}\mathbb{E}_{\mathcal{O}}[G_o^2] - G(q)^2. \tag{26}$$

Note that $\frac{|\mathcal{S}_t^o|(2|\mathcal{O}|-|\mathcal{S}_t^o|)}{|\mathcal{O}|^2} = \frac{2|\mathcal{S}_t^o|}{|\mathcal{O}|} - \frac{|\mathcal{S}_t^o|^2}{|\mathcal{O}|^2} = 1 - \left(1 - \frac{|\mathcal{S}_t^o|}{|\mathcal{O}|}\right)^2 \leq 1.$

Therefore:

$$\boxed{\mathrm{Var}[\gamma G_o] \leq \mathbb{E}_{\mathcal{O}}[G_o^2] - G(q)^2 = \mathrm{Var}_{\mathcal{O}}[G_o].} \tag{27}$$

i.e., the completion-level variance does not increase.

## D.2. Prompt-Level Variance

We now analyze the variance over prompts, where $G(q) = \mathbb{E}_o[\Psi(q,o)]$ and $\bar{G} = \mathbb{E}_q[G(q)]$ denotes the grand mean. With $C_q = \frac{|\mathcal{S}_t^q|}{|\mathcal{Q}|}$, by the same analysis:

$$\mathrm{Var}[\gamma(q)G(q)] = \frac{|\mathcal{S}_t^q|}{|\mathcal{Q}|^2}\sum_{q\in\mathcal{Q}}\frac{G(q)^2}{1-\mathcal{P}_t^q(q)} - \bar{G}^2. \tag{28}$$

Similarly,

$$\mathbb{E}_{q|\mathcal{P}_t^q(q)\neq 0}\left[\frac{G(q)^2}{1-\mathcal{P}_t^q(q)}\right] \leq \mathbb{E}_{\mathcal{Q}}[G(q)^2], \tag{29}$$

we obtain:

$$\boxed{\mathrm{Var}[\gamma(q)G(q)] \leq \mathrm{Var}_{\mathcal{Q}}[G(q)].} \tag{30}$$

## D.3. Total Variance

The total variance of $\hat{G} = \gamma(q)\gamma(o,q)\Psi(q,o)$ decomposes via the law of total variance:

$$\mathrm{Var}[\hat{G}] = \underbrace{\mathrm{Var}_q[\gamma(q)G(q)]}_{\text{Term I}} + \underbrace{\mathbb{E}_{q\sim\tilde{P}}[\gamma(q)^2\mathrm{Var}_o[\gamma(o,q)\Psi]]}_{\text{Term II}}. \tag{31}$$

**Term I:** From (30), $\mathrm{Var}_q[\gamma(q)G(q)] \leq \mathrm{Var}_{\mathcal{Q}}[G(q)].$

**Term II:** From (27), $\mathrm{Var}_o[\gamma(o,q)\Psi] \leq \mathrm{Var}_{\mathcal{O}}[\Psi].$ For $\mathbb{E}_{q\sim\tilde{P}}[\gamma(q)^2]$:

$$\mathbb{E}_{q\sim\tilde{P}}[\gamma(q)^2] = \frac{C_q}{|\mathcal{Q}|}\sum_q\frac{1}{1-\mathcal{P}_t^q(q)} = \frac{|\mathcal{S}_t^q|}{|\mathcal{Q}|^2}\sum_q\frac{1}{1-\mathcal{P}_t^q(q)}. \tag{32}$$

Under score-based pruning with low-score prompts (fraction $\beta$) pruned with probability $r_q$:

$$\mathbb{E}_{q \sim \tilde{P}}[\gamma(q)^2] = \frac{(1 - \beta r_q)\,[1 - (1 - \beta) r_q]}{1 - r_q}. \tag{33}$$

**Theorem D.1** (**Total Variance Bound**). *Under DPPO's hierarchical pruning mechanism, the gradient estimator variance is bounded by:*

$$\mathrm{Var}[\hat{G}] \leq \mathrm{Var}_{\mathcal{Q}}[G(q)] + \frac{(1 - \beta r_q)\,[1 - (1 - \beta) r_q]}{1 - r_q} \cdot \mathbb{E}_q[\mathrm{Var}_{\mathcal{O}}[\Psi]]. \tag{34}$$

*For moderate pruning rates (e.g., $r_q, r_o \leq 0.7$ with $\beta = 0.5$), $\frac{(1-\beta r_q)[1-(1-\beta)r_q]}{1-r_q} \leq 1.42$, and the total variance remains well-controlled. For aggressive pruning (e.g., $r_q = r_o = 0.9$), this factor increases to approximately $3.025$, leading to noticeably higher variance as confirmed in our experiments (Figure 5).*

The above analysis provides insight into the design of dynamic pruning methods with rescaling. Conditions (23) and (29) suggest an important criterion for variance-controlled pruning: pruning decisions should preferentially target samples whose rescaled gradient contributions do not exceed the population average, ensuring that variance remains bounded while achieving computational savings.

## E. Case Study

**Case Study.** To provide qualitative insights into how DPPO improves reasoning quality, we present a case study in Table 11. It shows a representative MATH example with completions generated by four RL variants: GRPO, GRESO, CPPO, and DPPO. This problem requires finding the maximum value of $x_1(x_2 + x_3 + \cdots + x_{101})$ subject to the constraint $\sum_{i=1}^{101} x_i^2 = 1$, which demands careful application of the Cauchy-Schwarz inequality with proper accounting of the 100 terms involved.

While all methods correctly identify Cauchy-Schwarz as the key technique, GRPO, GRESO, and CPPO fail to properly account for the number of variables, arriving at the incorrect answer $\frac{1}{2}$. In contrast, **DPPO (ours)** with aggressive pruning rates ($r_q = 0.9$, $r_o = 0.9$) is the *only* method that correctly recognizes the 100-term structure and computes the right answer of 5 (✓).

This qualitative difference aligns with our quantitative analysis in Section C: by preferentially retaining samples with high absolute advantage during training, DPPO focuses on challenging problems where the model exhibits high uncertainty. This emphasis on "learning frontier" samples enables the model to develop more robust reasoning capabilities for complex, multi-step problems—precisely the type of problem illustrated in this example. In addition to improved accuracy, DPPO achieves up to $2.37\times$ speedup on the MATH setting, demonstrating that our pruning-based optimization can simultaneously accelerate training and enhance reasoning quality.

*Table 11.* Comparisons of completions generated by four methods on questions sampled from the test set of MATH dataset. All responses are produced using the MATH-trained Qwen3-8B model. ✓ and ✗ denote correct and incorrect responses, respectively.

---

**Question:**
Let $x_1, x_2, \ldots, x_{101}$ be positive real numbers such that $x_1^2 + x_2^2 + \cdots + x_{101}^2 = 1$. Find the maximum value of

$$x_1 x_2 + x_1 x_3 + \cdots + x_1 x_{101}.$$

Let's think step by step and output the final answer within box.
**Ground Truth: 5**

---

**GRPO Response ✗:**
<think> Okay, so I need to find the maximum value of the sum $x_1 x_2 + x_1 x_3 + \cdots + x_1 x_{101}$, given that the sum of the squares of all the $x_i$'s is 1. Hmm, let me think. First, the expression we need to maximize is the sum of all products of $x_1$ with each of the other variables. So that's like $x_1$ multiplied by $(x_2 + x_3 + \cdots + x_{101})$. Let me write that down: Sum $= x_1(x_2 + x_3 + \cdots + x_{101})$. Let me denote $S = x_2 + x_3 + \cdots + x_{101}$. Then the sum becomes $x_1 S$. So maybe I can use some inequality here. Since we have a constraint on the sum of squares, maybe Cauchy–Schwarz or AM–QM or something? Wait, Cauchy–Schwarz might be useful here. Let me recall that for vectors $\mathbf{a}$ and $\mathbf{b}$, $(\mathbf{a} \cdot \mathbf{b})^2 \leq (\mathbf{a} \cdot \mathbf{a})(\mathbf{b} \cdot \mathbf{b})$. But here, maybe I can think of $x_1$ and $S$ as part of some vectors? Alternatively, maybe I can use the Cauchy–Schwarz inequality on $x_1$ and the vector $(x_2, x_3, \ldots, x_{101})$. Let me try that. The sum $x_1 S$ is the dot product of $(x_1, 0, 0, \ldots, 0)$ and $(x_2, x_3, \ldots, x_{101})$. Then by Cauchy–Schwarz: $x_1 S \leq \sqrt{x_1^2 + 0 + \cdots + 0} \ \sqrt{x_2^2 + x_3^2 + \cdots + x_{101}^2} = x_1 \sqrt{1 - x_1^2}$. So then the sum is $\leq x_1 \sqrt{1 - x_1^2}$. Now, to maximize this expression with respect to $x_1$, let me set $f(x) = x\sqrt{1 - x^2}$. To find its maximum, take derivative: $f'(x) = \sqrt{1 - x^2} + x \cdot \frac{1}{2\sqrt{1-x^2}} \cdot (-2x) = \frac{1 - 2x^2}{\sqrt{1-x^2}}$. Set derivative to zero: $1 - 2x^2 = 0 \Rightarrow x^2 = \frac{1}{2} \Rightarrow x = \frac{1}{\sqrt{2}}$. Therefore, the maximum value is $\frac{1}{\sqrt{2}}\sqrt{1 - \frac{1}{2}} = \frac{1}{\sqrt{2}} \cdot \frac{1}{\sqrt{2}} = \frac{1}{2}$. So the maximum sum is $\frac{1}{2}$. Therefore, the answer is $\boxed{\frac{1}{2}}$. Wait, but let me check if this is correct. Because when we apply Cauchy–Schwarz, we have that the maximum of $x_1 S$ is $\frac{1}{2}$, given that $x_1^2 + (\text{sum of squares of others}) = 1$. So yes, that seems right. Therefore, the maximum value is $\frac{1}{2}$. **Final Answer** $\boxed{\frac{1}{2}}$ </think>

---

**GRESO Response ✗:**
<think> Okay, so I need to find the maximum value of the sum $x_1 x_2 + x_1 x_3 + \cdots + x_1 x_{101}$ given that the sum of the squares of all the variables is 1. Let me think about how to approach this. First, the problem is about maximizing a sum of products of variables, given a constraint on their squares. This seems like a problem where the Cauchy–Schwarz inequality or Lagrange multipliers might be useful. Since all variables are positive, maybe Cauchy–Schwarz is the way to go. Let me write down the expression I need to maximize. The sum is $x_1(x_2 + x_3 + \cdots + x_{101})$. Let me denote $S = x_2 + x_3 + \cdots + x_{101}$. Then the expression becomes $x_1 S$. So I need to maximize $x_1 S$, given that $x_1^2 + x_2^2 + \cdots + x_{101}^2 = 1$. Hmm. Maybe I can use the Cauchy–Schwarz inequality on $x_1$ and $S$. Let me recall that for any vectors $\mathbf{a}$ and $\mathbf{b}$, $(\mathbf{a} \cdot \mathbf{b}) \leq \|\mathbf{a}\| \|\mathbf{b}\|$. If I take $\mathbf{a} = (x_1, 0, \ldots, 0)$ and $\mathbf{b} = (0, x_2, x_3, \ldots, x_{101})$, then their dot product is $x_1 S$. The norm of $\mathbf{a}$ is $x_1$, and the norm of $\mathbf{b}$ is $\sqrt{x_2^2 + \cdots + x_{101}^2}$. Therefore, $x_1 S \leq x_1 \sqrt{x_2^2 + \cdots + x_{101}^2}$. But then, the right-hand side is $x_1 \sqrt{1 - x_1^2}$. So we consider the maximum of $x_1 \sqrt{1 - x_1^2}$. Let $f(x) = x\sqrt{1 - x^2}$. Taking derivative: $f'(x) = \sqrt{1 - x^2} + x \cdot \frac{1}{2\sqrt{1-x^2}} \cdot (-2x) = \sqrt{1 - x^2} - \frac{x^2}{\sqrt{1-x^2}} = \frac{1 - 2x^2}{\sqrt{1-x^2}}$. Setting to zero gives $1 - 2x^2 = 0 \Rightarrow x = \frac{1}{\sqrt{2}}$. Then maximum value is $\left(\frac{1}{\sqrt{2}}\right)\sqrt{1 - \frac{1}{2}} = \left(\frac{1}{\sqrt{2}}\right)\left(\frac{1}{\sqrt{2}}\right) = \frac{1}{2}$. Therefore, the maximum value is $\frac{1}{2}$. But wait, is this tight? Because we have equality in Cauchy–Schwarz when vectors are proportional. So for equality, we need $(x_1, 0, \ldots, 0)$ and $(0, x_2, \ldots, x_{101})$ to be proportional, which would mean that $x_2 = x_3 = \cdots = x_{101}$. Let me check. If all $x_2$ to $x_{101}$ are equal, denote them as $y$. Then the sum of squares is $x_1^2 + 100y^2 = 1$. The expression to maximize is $x_1 \cdot 100y$. From the earlier result, maximum of $x_1 \cdot 100y$ is $100 \cdot \frac{1}{2} = 50$? Wait, no. Earlier we found that $x_1 \sqrt{1 - x_1^2}$ has maximum $\frac{1}{2}$, but here $\sqrt{1 - x_1^2} = \sqrt{100y^2} = 10y$. Therefore, $x_1 \cdot 10y \leq \frac{1}{2}$. Therefore, the maximum of $x_1 \cdot 10y$ is $\frac{1}{2}$. Therefore, the total sum $x_1(x_2 + \cdots + x_{101})$ is $100 x_1 y$, so $100 x_1 y \leq \frac{1}{2}$. Therefore, maximum value is $\frac{1}{2}$. Therefore, the answer is $\boxed{\frac{1}{2}}$. **Final Answer** $\boxed{\frac{1}{2}}$
</think>

**CPPO Response ✗:**

<think> Okay, so I need to find the maximum value of the sum $x_1x_2 + x_1x_3 + \cdots + x_1x_{101}$, given that the sum of the squares of all the $x_i$'s is 1. Let me think about how to approach this. First, the problem is about maximizing a sum of products involving $x_1$ with each of the other variables. The constraint is on the sum of squares. This seems like an optimization problem with a constraint, so maybe I can use Lagrange multipliers? Or perhaps there's a way to apply the Cauchy–Schwarz inequality or some other inequality? Let me write down what I need to maximize: $S = x_1x_2 + x_1x_3 + \cdots + x_1x_{101}$. So that's $x_1$ multiplied by the sum of $x_2$ through $x_{101}$. Let me denote that sum as $T = x_2 + x_3 + \cdots + x_{101}$. Then $S = x_1T$. So the problem becomes maximizing $x_1T$, given that $x_1^2 + x_2^2 + \cdots + x_{101}^2 = 1$. Hmm. Maybe I can express $T$ in terms of the other variables? Or maybe use Cauchy–Schwarz on $T$? Alternatively, think of $S$ as $x_1\left(\sum_{i=2}^{101} x_i\right)$. Let me consider variables $x_2, x_3, \ldots, x_{101}$. Let me denote them as $y_1, y_2, \ldots, y_{100}$, so that $T = y_1 + y_2 + \cdots + y_{100}$. Then the constraint becomes $x_1^2 + y_1^2 + y_2^2 + \cdots + y_{100}^2 = 1$. So we need to maximize $x_1(y_1 + \cdots + y_{100})$ with the constraint $x_1^2 + \sum y_i^2 = 1$. This seems like a standard problem. Maybe use Cauchy–Schwarz on the terms $x_1$ and the sum of $y_i$? Wait, let me think. Let me consider vectors. Suppose I have two vectors: one is $(x_1, y_1, y_2, \ldots, y_{100})$ and another... Hmm, maybe not. Alternatively, think of the sum $x_1(y_1 + \cdots + y_{100})$ as the dot product of $(x_1, 0, 0, \ldots, 0)$ and $(y_1, y_2, \ldots, y_{100})$. But I don't know if that helps directly. Alternatively, use Cauchy–Schwarz on the terms $x_1$ and the sum of $y_i$. Let me recall that for any vectors $\mathbf{u}$ and $\mathbf{v}$, $|\mathbf{u} \cdot \mathbf{v}| \leq \|\mathbf{u}\|\,\|\mathbf{v}\|$. If I take $\mathbf{u} = (x_1, 0, 0, \ldots, 0)$ and $\mathbf{v} = (y_1, y_2, \ldots, y_{100})$, then their dot product is $x_1(y_1 + \cdots + y_{100}) = S$. The norm of $\mathbf{u}$ is $\sqrt{x_1^2} = x_1$. The norm of $\mathbf{v}$ is $\sqrt{y_1^2 + \cdots + y_{100}^2}$. Therefore, by Cauchy–Schwarz: $S \leq x_1\sqrt{y_1^2 + \cdots + y_{100}^2}$. But the constraint is $x_1^2 + y_1^2 + \cdots + y_{100}^2 = 1$. Let me denote that sum of squares of $y_i$ as $Q$. Then $Q = 1 - x_1^2$. Therefore: $S \leq x_1\sqrt{1 - x_1^2}$. So now, the problem reduces to maximizing $x_1\sqrt{1 - x_1^2}$ over $x_1 > 0$. Let me set $f(x) = x\sqrt{1 - x^2}$. To find its maximum, take derivative: $f'(x) = \sqrt{1 - x^2} + x\left(\frac{1}{2\sqrt{1-x^2}}\right)(-2x) = \sqrt{1 - x^2} - \frac{x^2}{\sqrt{1-x^2}} = \frac{(1-x^2)-x^2}{\sqrt{1-x^2}} = \frac{1-2x^2}{\sqrt{1-x^2}}$. Set derivative equal to zero: $1 - 2x^2 = 0 \Rightarrow x^2 = \frac{1}{2} \Rightarrow x = \frac{1}{\sqrt{2}}$. Therefore, maximum value is $f\left(\frac{1}{\sqrt{2}}\right) = \left(\frac{1}{\sqrt{2}}\right)\sqrt{1 - \frac{1}{2}} = \left(\frac{1}{\sqrt{2}}\right)\left(\frac{1}{\sqrt{2}}\right) = \frac{1}{2}$. Therefore, the answer is $\boxed{\frac{1}{2}}$. **Final Answer** $\boxed{\frac{1}{2}}$ </think>

**DPPO Response (Ours) ✓:**

<think> Okay, so I need to find the maximum value of the sum $x_1x_2 + x_1x_3 + \cdots + x_1x_{101}$, given that the sum of the squares of $x_1$ to $x_{101}$ is 1. All the $x_i$ are positive real numbers. Hmm, let me think about how to approach this. First, the expression we need to maximize is the sum of all products of $x_1$ with each of the other variables. So that's $x_1$ multiplied by $(x_2 + x_3 + \cdots + x_{101})$. Let me denote $S = x_2 + x_3 + \cdots + x_{101}$. Then the expression becomes $x_1 \cdot S$. Now, we know from the given condition that $x_1^2 + x_2^2 + \cdots + x_{101}^2 = 1$. So maybe I can relate $S$ and $x_1$ using some inequality. Since all variables are positive, perhaps Cauchy-Schwarz inequality could be useful here. **By Cauchy-Schwarz,** $(x_2^2 + x_3^2 + \cdots + x_{101}^2)(1^2 + 1^2 + \cdots + 1^2) \geq (x_2 + x_3 + \cdots + x_{101})^2$. There are 100 terms in the sum for $x_2^2$ to $x_{101}^2$, so that's 100 terms. **Therefore,** $(\sum_{i=2}^{101} x_i^2) \cdot 100 \geq S^2$. Hence, $S^2 \leq 100 * (1 - x_1^2)$. Therefore, $S \leq 10 * \sqrt{1 - x_1^2}$. So our expression $x_1 * S \leq x_1 * 10 * \sqrt{(1 - x_1^2)}$. Now, we need to maximize this expression with respect to $x_1$. Let me set $f(x) = 10x * \sqrt{(1 - x^2)}$. To find its maximum, take derivative: $f'(x) = 10 * \sqrt{(1-x^2)} + 10x * (1/(2*\sqrt{(1-x^2)}))*(-2x) = 10 * \sqrt{(1-x^2)} - 10x^2/\sqrt{(1-x^2)} = [10(1-x^2) - 10x^2]/\sqrt{(1-x^2)} = 10(1-2x^2)/\sqrt{(1-x^2)}$. Set derivative equal to zero: $1 - 2x^2 = 0 \Rightarrow x^2 = 1/2 \Rightarrow x = 1/\sqrt{2}$. Since $x$ is positive. So maximum value is $10 * (1/\sqrt{2}) * \sqrt{(1 - 1/2)} = 10 * (1/\sqrt{2}) * (1/\sqrt{2}) = 10 * (1/2) = 5$. Therefore, the maximum value is 5. So the answer is $\boxed{5}$. **Final Answer** $\boxed{5}$ </think>

