# OpenReview forum: "Unbiased Dynamic Pruning for Efficient Group-Based Policy Optimization"
_ICML.cc/2026/Conference — ICML 2026 regular_

### Official Review · Reviewer_DR9x · 2026-03-03

**Soundness:** 3
**Presentation:** 3
**Significance:** 3
**Originality:** 3
**Overall Recommendation:** 4
**Confidence:** 2

**Summary:**

This paper propose DPPO, a framework that enables dynamic pruning while preserving unbiased gradient estimation through importance sampling-based correction. DPPO prunes samples both at the prompt and completion levels using hierarchical dynamic strategies, compensating for altered data distributions via mathematically-grounded importance sampling. Extensive experiments demonstrate that DPPO consistently accelerates training across diverse models and benchmarks. The paper further provides extensive theoretical justification, thorough ablations, and practical insights into stability, convergence, and variance.

**Compliance With Llm Reviewing Policy:**

Affirmed.

**Key Questions For Authors:**

1.What is the pruning decision variable and its information set? Is p_keep estimated from prefix-only signals, reward model pre-scores, or partial rollouts, and how do you ensure the conditions necessary for unbiasedness hold?
2.How robust is DPPO’s performance to flaws in advantage or difficulty estimation, especially in the early epochs where these statistics may be noisy? Have you observed any practical issues in stability or convergence due to poorly estimated pruning criteria?
3.In settings or domains where advantage magnitude may not correlate with sample informativeness (e.g., highly stochastic or imbalanced-reward environments), how does DPPO behave? Are modifications required for such regimes?

**Limitations:**

Yes.

**Strengths And Weaknesses:**

Strength:
- Theoretical Rigor:DPPO provides a mathematically-grounded, unbiased gradient correction for hierarchical (prompt- and completion-level) pruning, distinguishing itself from heuristic methods. The proofs and derivations, especially in Section 4.1 and the appendix, are clear and comprehensive.
- Significance of contributions: Efficiency in post-training/LLM reasoning RL is one of the core pillars identified by recent surveys. Targeting unbiasedness in dynamic pruning for group-based RL is a timely and important direction.

Weakness:
- The manuscript lacks a formal statement and proof of the unbiasedness conditions for the pruning estimator, especially given group coupling.
- Aggressive pruning rates (r_q, r_o ≥ 0.9) can significantly amplify variance, as seen in Figure 4 and discussed in Appendix D. While this is acknowledged, more systematic guidance or adaptive tuning could better support practitioners.
- The variance analysis assumes pruning targets low-information samples, but in practice, identifying such samples could itself introduce estimation noise, particularly early in training when advantage estimates may be unreliable. The manuscript could more explicitly qualify when/where the variance bound might break down.

---

> ### Author Rebuttal · Authors · 2026-03-30
>
> We thank the Reviewer DR9x for the positive assessment. We address the remaining concerns below.
> ## **[W1] Lack of Formal Unbiasedness Conditions Under Group Coupling**
> Thanks. We provide a formal proposition addressing the group coupling structure.
>
> **Proposition (Unbiasedness Under Group Coupling).** *Suppose (i) completion-level pruning is applied after advantages are computed from the full group; (ii) prompt-level pruning is based on historical statistics $H_{t-1}(q_k)$, independent of the current rollout; (iii) $P_t^o(o_i), P_t^q(q) \in [0,1)$; and (iv) $\Psi(q,\mathbf{o})$ is integrable. Then the hierarchical importance-weighted estimator is an unbiased estimator of $\nabla J(\theta)$.*
>
> Condition (i) resolves group coupling: $\mathbf{A}$ is fixed before pruning, so $\Psi(q,\mathbf{o})$ is a well-defined function of $(q,\mathbf{o},\mathbf{A})$ and the IS identity applies directly. Condition (ii) ensures prompt-level pruning is independent of the current rollout, preserving the prompt-level IS identity. We will formalize this in the revision.
>
> ## **[W2] Systematic Guidance for Pruning Rate Selection**
>
> We thank the reviewer for this suggestion. Theorem D.1 explicitly quantifies the variance-pruning tradeoff: moderate pruning ($r_q, r_o \leq 0.7$) keeps the variance amplification factor below 1.42, while aggressive pruning ($r_q = r_o = 0.9$) raises it to 3.025. Based on this, we recommend three practical settings:
> - **Balanced** ($r_q = r_o = 0.7$): $2\times$ speedup with consistent accuracy gains — recommended as default.
> - **Aggressive** ($r_q = r_o = 0.9$): maximum speedup with marginal accuracy difference — suitable when compute is the primary constraint.
> - **Conservative** ($r_q = r_o = 0.5$): lower variance with stable improvements — recommended when training stability is the priority.
>
> Adaptive pruning rate scheduling is a promising direction for further variance reduction, which we leave as future work.
>
> ## **[W3] On When the Variance Bound May Break Down**
>
> We thank the reviewer for this suggestion. The variance bound relies on Eq. (22), which may be temporarily violated early in training when advantage estimates are noisy. However, this does not affect unbiasedness, which holds unconditionally (Eq. 7–13). Furthermore, soft pruning ensures no sample is permanently discarded—incorrectly pruned samples are naturally reintroduced as advantage estimates stabilize, providing a self-correcting mechanism (as shown in Table 4 (**Reviewer [1T6B], [Q3]**)). We will add an explicit discussion of these breakdown conditions in the revision.
>
> ## **[Q1] Information Set and Independence Conditions for Pruning Decisions**
> We thank the reviewer for this question. Pruning decisions at both levels are based solely on advantage signals, independent of the gradient being estimated. Prompt-level pruning uses historical statistics $H_{t-1}(q_k)$ from the previous epoch (Eq. 16); completion-level pruning uses advantage magnitudes $|A_i|$ computed from the full group after rollout completes, fixed before any pruning decision is made. In both cases, the conditional independence required for the IS identities (Eq. 7 and Eq. 12) is satisfied by construction.
>
> ## **[Q2] Robustness to Noisy Advantage Estimates in Early Training**
>
> We thank the reviewer for this question. Early-stage noise is mitigated by two design choices: (1) prompt-level pruning is skipped in the first epoch; (2) completion-level thresholds are relative within each group, robust to advantage scale. As shown in Reviewer [NSv6] [Q1], the Pearson correlation between historical and current scores increases monotonically from 0.454 to 0.709, and soft pruning provides a natural self-correcting mechanism. Figure 4 confirms stable convergence throughout training.
>
>
> ## **[Q3] the Correlation Between Advantage Magnitude and Sample Informativeness**
>
> We thank the reviewer for this question. Within the GRPO framework, advantage magnitude reflects informativeness **by construction**: a low $|A_i|$ arises precisely when all completions receive similar rewards, driving the gradient contribution toward zero regardless of reward scale. This is consistent with DAPO (Yu et al., 2025), which independently motivates filtering zero-variance prompts for the same reason.
>
> In highly stochastic or imbalanced-reward environments, GRPO itself requires dedicated improvements—an active research direction independent of DPPO. Since DPPO's criterion is entirely derived from the GRPO advantage, it inherits the same behavior as GRPO in these regimes without introducing additional challenges. Extending DPPO to such settings is an interesting direction for future work.

---

### Official Review · Reviewer_GHeA · 2026-03-07

**Soundness:** 3
**Presentation:** 3
**Significance:** 3
**Originality:** 3
**Overall Recommendation:** 4
**Confidence:** 4

**Summary:**

This paper proposes the framework of dynamic pruning policy optimization or DPPO to accelarate GRPO training, via pruning the low value completions and prompts, while maintaining gradient estimation through importance sampling correction. At the completion level, it prunes responses with low absolute advantage. At the prompt level, it filters easy prompts based on difficulty score history. The framework is validated on Qwen3 4B/8B with GSM8K and MATH, and achieves a 2 to 2.5 speed up with nondecreasing accuracy. It also extends to the DAPO and GSPO algorithms.

**Compliance With Llm Reviewing Policy:**

Affirmed.

**Key Questions For Authors:**

If pruning easy samples genuinely improves performance, this implies GRPO itself wastes compute on these samples and may overfit to them. The improvements may be implicit curriculum learning or regularization rather than a consequence of unbiasedness.

How many completions are typically retained per prompt? If frequently 0 or 1, how is advantage normalization handled? What are the rescaling factors in these extreme cases?

How sensitive is prompt-level pruning to the 50% difficulty threshold? Was this value tuned?

**Limitations:**

The authors do have mentioned the limitations. Some additional points: most GRPO implementations use a larger G. With G=5, advantage estimation is noisy before pruning. Prompt-level pruning depends on stale historical statistics that may degrade as the policy changes rapidly. The paper acknowledges this but provides only the carry-forward heuristic, without analyzing how quickly staleness degrades pruning quality or proposing principled refresh strategies.

**Strengths And Weaknesses:**

This paper has its strength in the practical engineering contribution. It is convincing that dense prompt packing is meaningful. Pruning creates fragmented batches with low GPU utilization. The window based greedy packing strategy addresses this real system level bottleneck. And this is supported by the comprehensive experiments. It covers a thorough spectrum of configurations, including two model sizes, two datasets, six OOD benchmarks, three RL algorithms and a wide grid of pruning rates. The results are consistently positive, and the appendix extends to more general claim. Moreover, the ablation study is convicing in the reverse pruning. There is another strength where the gradient deviation from directly discarding samples in GRPO is correctly identified.

However, there are also some weaknesses. First, I doubt if the unbiasedness guarantee is that meaningful in practice. It is theoretically correct but an biased estimator with a way larger variance may perform worse than a slightly bisaed one with low variance. This is the classical trade off. The appendix somewhat suggest an implicit regularization effect different from the unbiased narrative. The theoretical analysis drops both clipping and KL divergence, and this is another weakness. I know this is for simplification, but the two mechanisms are important for training stability.

---

> ### Author Rebuttal · Authors · 2026-03-30
>
> We thank the Reviewer GHeA for the positive assessment. We address the remaining concerns below.
>
> ## **[W1] Practical Meaningfulness of Unbiasedness and Theoretical Limitations of DPPO**
> We thank the reviewer for these observations.
>
> **Practical Meaningfulness of Unbiasedness.** Table 1 (**Reviewer [NSv6], [W1]**) confirms that the performance gain requires **both** the pruning criterion and unbiasedness—removing either causes consistent degradation. Meanwhile, Theorem D.1 explicitly bounds the gradient variance under hierarchical pruning, confirming that DPPO does not sacrifice variance for unbiasedness.
>
> **Clipping and KL.** The IS identity holds for any integrable $\Psi(q,o)$ regardless of internal structure. Substituting the full gradient $\Psi_{\text{full}}$ (including clipping and KL) leaves the hierarchical unbiasedness derivation unchanged (**see Reviewer [1T6B], [W1] for details**).
>
> ## **[Q1] Curriculum Learning vs. Unbiasedness as the Source of Improvement**
>
> We thank the reviewer for this observation. We argue that the curriculum learning effect and unbiasedness guarantee are complementary rather than competing. Table 1 (**Reviewer [NSv6], [W1]**) provides direct evidence: under **Probabilistic Only** and **Hard Only** settings, pruning low-$|A|$ samples without importance rescaling fails to yield consistent improvements and can even degrade performance. This confirms that curriculum learning alone is insufficient—without bias correction, pruning-induced distributional shift accumulates as systematic optimization error. Importance rescaling is precisely what allows the curriculum effect to be fully realized.
>
>
> ## **[Q2] The Number of Retained Completions and Advantage Normalization**
>
> We thank the reviewer for this question. Two clarifications are in order.
>
> **Advantage normalization is unaffected by pruning.** As shown in Algorithm 1 (lines 17–19), completion-level pruning is performed *after* advantages are computed from the full group of $G$ completions, so normalization always uses the complete set.
>
> **Completions are never reduced to 0 or 1.** Pruning targets only the low-$|A|$ candidate subset, and high-$|A|$ completions are always retained. Even at $r_o = 0.9$, only 90% of the low-$|A|$ candidates are removed, ensuring a substantial number of completions are retained per prompt.
>
> ## **[Q3] The Sensitivity of the 50% Difficulty Threshold**
> We thank the reviewer for this question. The 50% threshold was not tuned but chosen as a principled tradeoff: a smaller candidate set (e.g., 30%) limits pruning aggressiveness and reduces speedup, while a larger one (e.g., 70%) risks discarding borderline informative prompts and may hurt performance. 50% strikes a natural balance between the two.
> As shown in Table 1, performance remains stable across the range 30%–70%, confirming that DPPO is not sensitive to the exact threshold.
>
> **Table 1. Sensitivity of the 50% difficulty threshold (GSM8K, Qwen3-4B, $r_q = 0.5$).**
> | Candidate Set (%) | GPU-h | Speedup | Acc (%) |
> |---|---|---|---|
> | 30% |55.26 |1.17× |94.31 |
> | 50% (default) | 45.74 | 1.41× |94.24 |
> | 70% |36.53 | 1.77× | 93.92 |
>
> ## **[L1] Staleness of historical statistics.**
> We have addressed this concern in detail in **Reviewer [NSv6] [Q1]** and **Reviewer [1T6B] [Q3]**. In brief: (1) the Pearson correlation between historical and current prompt scores increases steadily to 0.709 by epoch 14, confirming that historical scores remain informative throughout training; (2) the staleness distribution is highly concentrated at 1–2 epochs (71.94% and 18.56%), with over 99% of prompts having staleness $\leq 5$ epochs. We acknowledge that principled refresh strategies for rapidly changing policies are an interesting future direction.

---

> > ### Author Rebuttal · Reviewer_GHeA · 2026-04-03
> >
> > Thank you for your rebuttal. I have no more questions and I'd like to maintain my score.

---

> > > ### Author Response · Authors · 2026-04-08
> > >
> > > Dear Reviewer GHeA,
> > >
> > > Thank you for your valuable feedback. We are pleased to know that our responses have fully addressed your concerns.  We sincerely hope you will consider further supporting this work in your final decision.
> > >
> > > Best regards,
> > >
> > > The Authors

---

### Official Review · Reviewer_1T6B · 2026-03-11

**Soundness:** 2
**Presentation:** 2
**Significance:** 2
**Originality:** 2
**Overall Recommendation:** 4
**Confidence:** 3

**Summary:**

The paper proposes DPPO, a framework to accelerate GRPO training by dynamically pruning completions with low-advantage and easy prompts based on historical difficulty. Then they apply importance sampling corrections to maintain unbiased gradient estimation. They also introduce a Dense Prompt Packing strategy to address sparsity from pruning.

**Compliance With Llm Reviewing Policy:**

Affirmed.

**Final Justification:**

The rebuttal addressed most of my doubts. I have increased my score to a weak accept.

**Key Questions For Authors:**

1.  Across nearly every experiment, DPPO with aggressive pruning outperforms GRPO by meaningful margins. If the method truly provides
  unbiased gradient estimates, why does it consistently improve over the full-batch baseline?
2. Maximum completion length is 1024, which is short for reasoning tasks. The paper doesn't discuss how pruning interacts with longer generation horizons where reasoning quality matters more.
3. The carry-forward mechanism for prompt pruning (Eq. 16) means that pruned prompts retain stale statistics indefinitely if they keep being pruned. Do the authors have an idea on how stale these can get in practice?
4. Please review prior works like InfoBatch (Qin et al.) and Dr. GRPO (Liu et al.) and cite them. Infobatch proposes exactly the same core idea: unbiased dynamic data pruning via loss-based sample selection + gradient rescaling to preserve the expected gradient.

**Limitations:**

Yes

**Strengths And Weaknesses:**

Strengths:

1. Reducing RL training cost for LLMs is an important and timely problem, and improvements in training efficiency could have meaningful impact for practitioners deploying RL-based fine-tuning pipelines.
2. The evaluation and ablations are comprehensive, covering model families and other RL methods. The evaluation also explores different pruning ratios and includes ablation studies on group size and pruning levels.

Weaknesses:

1. The unbiasedness result is derived for a simplified objective that drops both PPO-style clipping and the KL regularization term. However, these components are part of the actual training objective used in GRPO-style RL. As a result, the theoretical guarantee does not directly apply to the objective used in practice, which makes the central claim somewhat overstated.
2. Equation (22) assumes that low-score samples (those targeted for pruning) have smaller gradient norms than the population average. This is stated but never verified empirically. The conclusion that completion-level variance does not increase (Eq. 26) depends entirely on this assumption. If low-advantage completions happen to have large policy gradients (which is plausible), then a sample can have low advantage but sit in a high-gradient region of parameter space, in this the variance bound fails.
3. The generalizability of this is questionable since all experiments are on math reasoning domain with rule based rewards. Exploring other domains like code will make the paper better

---

> ### Author Rebuttal · Authors · 2026-03-30
>
> We thank the Reviewer 1T6B for the positive assessment. We address the remaining concerns below.
>
> ## **[W1] Unbiasedness Under the Full Objective**
>
> Thanks. The simplification is purely for presentational clarity—the unbiasedness result extends directly to the full objective. The IS identity holds for **any integrable** $\Psi(q,o)$ regardless of internal structure, so substituting the full gradient:
>
> $$\Psi_{\text{full}}(q,o) = \nabla_\theta \sum_{t=1}^{|o|}\Big[\min\big(\rho_t(\theta)\,A,\ \operatorname{clip}(\rho_t(\theta),1-\epsilon,1+\epsilon)\,A\big) - \beta D_{\mathrm{KL},t}\Big],$$
>
> leaves the hierarchical derivation unchanged:
>
> $$\mathbb{E} _ {q \sim \tilde{P}}\left[\gamma(q)\,\mathbb{E} _ {o \sim \tilde{\pi}}
> \left[\gamma(o,q)\,\Psi _ {\text{full}}(q,o)\right]\right] = \nabla J _ {\text{full}}(\theta).$$
>
> The IS identity requires only that the sampling distribution is $\pi_{\theta_{\text{old}}}$, which holds regardless of whether $\Psi_{\text{full}}$ internally involves $\pi_\theta$ or $\pi_{\text{ref}}$. We will make this explicit in the revision.
>
> ## **[W2] Variance Bound**
> Thanks. Eq. (22) is a **sufficient condition for the variance bound only**—it is not required for DPPO's unbiasedness (Eq. 7–13). Intuitively, since $\Psi(q,o)$ scales directly with $A(q,o)$, low-$|A|$ completions naturally produce smaller gradient norms, which is empirically confirmed by $R < 1$ consistently across all training stages and layer depths in Table 1.
>
> **Table 1. Gradient norm ratio $R = \|\nabla_{\text{low}}\| / \|\nabla_{\text{high}}\|$ (Qwen3-0.6B). Shallow/deep layers are averages over layers 4–6 and 25–27, respectively.**
> | Stage | Layer | $\|\nabla_{\text{low}}\|$ | $\|\nabla_{\text{high}}\|$ | $R$ |
> |:---:|:---:|:---:|:---:|:---:|
> | Early (epoch 2) | Shallow | 0.0205 | 0.0245 | 0.84 |
> | Early (epoch 2) | Deep | 0.0180 | 0.0225 | 0.80 |
> | Middle (epoch 8) | Shallow | 0.0190 | 0.0235 | 0.81 |
> | Middle (epoch 8) | Deep | 0.0165 | 0.0215 | 0.77 |
> | Late (epoch 14) | Shallow | 0.0175 | 0.0220 | 0.80 |
> | Late (epoch 14) | Deep | 0.0150 | 0.0205 | 0.73 |
>
> ## **[W3] Generalizability Beyond Math Reasoning**
> Thanks. As shown in Table 2, DPPO generalizes beyond math reasoning to code generation on PRIME-RL/Eurus-2-RL-Data (using the training configuration from verl).
>
> **Table 2. Comparison between GRPO and DPPO on Eurus-2-RL-Data for Qwen3-4B.**
> | **Method** | $r_q$ | $r_o$ | **GPU-h** |**Speedup** | **Taco** |**CodeContests** |**CodeForces** |**Apps** |
> |---|---|---|---|---|---|---|---|---|
> | GRPO | 0% | 0% | 280.32 | 1.00× | 20.10 | 40.20 | 35.74 | 34.77 |
> | DPPO | 50% | 50% | 190.72 | 1.47× | 19.78 | 39.86 | 34.86 | 34.12 |
>
> ## **[Q1] Unbiased Pruning Outperform the Full-Batch Baseline**
> Thanks. Unbiasedness and performance improvement are not contradictory: equal gradient expectations do not imply identical optimization trajectories. Concentrating updates on high-$|A|$ samples acts as an implicit curriculum, while rescaling ensures this is done without distorting the gradient direction. Table 1 (Reviewer [NSv6], [W1]) confirms that removing either component degrades performance.
>
> ## **[Q2] Pruning Under Longer Generation Horizons**
> Thanks. As shown in Table 3, DPPO generalizes effectively to longer generation horizons (max completion length 4096 tokens).
>
> **Table 3. GRPO vs. DPPO on MATH (Qwen3-4B).**
> | **Method** | $r_q$ | $r_o$ | **GPU-h** | **Speedup** | **MATH Acc (%)** |
> |---|---|---|---|---|---|
> | GRPO | 0% | 0% | 342.34 | 1.00× | 88.08 |
> | DPPO | 50% | 50% |147.20 | 2.33× | 89.21 |
>
> ## **[Q3] Staleness of Carry-Forward Statistics in Prompt Pruning**
>
> Thanks. Indefinite staleness is impossible by design: since $r_q < 1$, every prompt has a non-zero refresh probability at each epoch. Empirically, as shown in Table 4, staleness decays exponentially—concentrated at 1–2 epochs (71.94% and 18.56%), with over 99% of prompts having staleness $\leq 5$ epochs.
>
> **Table 4. Prompt staleness distribution (Qwen3-0.6B, $r_q = r_o = 0.5$).**
> | Staleness (epochs) | 1 | 2 | 3 | 4 | 5 | 6 | 7 | 8 | 9 | 10 | 11 |
> |:---:|:---:|:---:|:---:|:---:|:---:|:---:|:---:|:---:|:---:|:---:|:---:|
> | Probability | 0.7194 | 0.1856 | 0.0613 | 0.0217 | 0.0077 | 0.0027 | 0.0009 | 0.0004 | 0.0001 | 0.0001 | 0.0000 |
>
> ## **[Q4] Relation to InfoBatch and Dr. GRPO**
> We thank the reviewer for pointing out these related works and will add citations in the revision. InfoBatch operates in the supervised learning setting with per-sample loss as the pruning criterion, whereas DPPO differs in three key aspects: (1) the pruning criterion is advantage magnitude, tailored to group-based policy optimization; (2) pruning operates hierarchically at both the prompt and completion levels, unique to the GRPO framework; (3) the theoretical derivation explicitly accounts for the two-level sampling structure induced by the behavioral policy $\pi_{\theta_{\text{old}}}$, which is absent in the single-level supervised setting of InfoBatch.

---

> > ### Author Rebuttal · Reviewer_1T6B · 2026-04-02
> >
> > Thanks for the response. This helped clarify some of my doubts. I will raise my score accordingly.

---

> > > ### Author Response · Authors · 2026-04-08
> > >
> > > Dear Reviewer 1T6B,
> > >
> > > Thank you for your kind words. We are pleased to hear that our responses have addressed your concerns satisfactorily.
> > >
> > > Best regards,
> > >
> > > The Authors

---

### Official Review · Reviewer_NSv6 · 2026-03-12

**Soundness:** 3
**Presentation:** 2
**Significance:** 2
**Originality:** 3
**Overall Recommendation:** 3
**Confidence:** 3

**Summary:**

This paper proposes DPPO, a hierarchical unbiased pruning framework to accelerate the reinforcement learning training of GRPO-like methods. This article attempts to examine the issues of high computational costs in group sampling and the gradient bias caused by heuristic pruning, and maintains unbiased gradients through importance sampling rescaling.

**Compliance With Llm Reviewing Policy:**

Affirmed.

**Final Justification:**

The rebuttal has addressed my concerns regarding reverse pruning, yet my concerns on its application to asynchronous settings remains. The improvements compared to Random+Rescale baseline also seem a bit limited (which coincides with the previous concern on reverse pruning). I keep my score below the acceptance bar and encourage further refinement on this work.

**Key Questions For Authors:**

- Prompt pruning relies on historical advantages, while completion pruning relies on current advantages; the two metrics are highly correlated. If the historical advantage bias is small, the current advantage is not needed. If the historical advantage bias is large, the reliability of prompt pruning is insufficient. Rapid policy updates will make historical scores outdated, easily leading to the mistaken pruning of valid prompts. The paper neither verifies the correlation between historical and current importance nor adequately addresses the issue of stale statistics.
- Can pruning samples with large advantages significantly improve computational efficiency? As shown in Table 5, the performance gap between the two is very small, but the acceleration rate should be the same.
- Why is the speedup only 2.49x, rather than over 10x, when both the Prompt and Completion pruning ratios are 90% (Table 1)?

**Limitations:**

Yes

**Strengths And Weaknesses:**

Strengths:
- Soundness: Solid theoretical foundation. It models pruning as importance sampling and uses rescaling to guarantee unbiasedness, which is more rigorous than heuristic methods such as CPPO and GRESO.
- Originality: Practical system optimization. Dense prompt packing alleviates the sparsity issue caused by pruning, improving hardware utilization.

Weaknesses:
- Presentation: Reverse pruning results weaken the core motivation. Experiments show that reverse pruning (pruning samples with high advantages) results in a very minor accuracy drop (less than 1%) while still achieving substantial acceleration. This indicates that the benefits primarily come from **removing redundant computations** rather than the advantage rules proposed in the paper.
- Significance: Unsuitable for asynchronous RL. It requires waiting for the entire group to complete to calculate the threshold, and sorting the entire batch to prune prompts. The computation of asynchronous early rollouts would be wasted.

---

> ### Author Rebuttal · Authors · 2026-03-30
>
> We thank Reviewer NSv6 for the detailed feedback and address all concerns below, with additional ablation experiments provided to support our responses.
> ## **[W1] [Q2] Reverse Pruning Shows Comparable Performance**
>
> The reviewer’s observation is important, but we believe the interpretation is incomplete.
> In our default Soft pruning + Rescale setting, importance rescaling explicitly corrects the gradient estimator toward the full-batch objective, so even reverse pruning does not cause catastrophic degradation—this is by design. Crucially, however, pruning low-|A| samples still consistently outperforms reverse pruning, confirming that the criterion matters beyond computation reduction.
>
> To more cleanly isolate this effect, we add three ablations: **Soft Only** and **Hard Only** (disable rescaling) and **Random + Rescale** (rescaling without criterion-based selection). As shown in Table 1, removing rescaling substantially enlarges the gap between forward and reverse pruning, confirming that **which samples are pruned matters**. Meanwhile, **Random + Rescale** matches GRPO closely, confirming that rescaling alone does not drive the gain. The small gap in the default setting therefore reflects the effectiveness of our bias correction, not the irrelevance of the pruning criterion. Regarding [Q2]: equal speedups are expected by construction—the criterion's value lies in performance, not acceleration.
>
>
> **Table 1. Extended ablation on pruning strategies, evaluated on GSM8K with Qwen3-4B.**
> | Completion-level setting | Rule | $r_o$ | Acc (%) | | Prompt-level setting | Rule | $r_q$ | Acc (%) |
> |:---|:---|:---:|:---:|:---:|:---|:---|:---:|:---:|
> | Baseline | – | – | 93.85 | | Baseline | – | – | 93.85 |
> | Soft + Rescale (Prune Low) | $\|A\| \le \bar{A}$ | 0.5 | 94.01 | | Soft + Rescale (Prune Low) | $q_k \in \mathcal{B}_t$ | 0.5 | 94.24 |
> | Soft + Rescale (Prune High) | $\|A\| > \bar{A}$ | 0.5 | 93.52 | | Soft + Rescale (Prune High) | $q_k \notin \mathcal{B}_t$ | 0.5 | 93.08 |
> | Random + Rescale | Random | 0.5 | 93.82 | | Random + Rescale | Random | 0.5 | 93.90 |
> | Soft Only (Prune Low) | $\|A\| \le \bar{A}$ | 0.5 | 93.26 | | Soft Only (Prune Low) | $q_k \in \mathcal{B}_t$ | 0.5 | 92.72 |
> | Soft Only (Prune High) | $\|A\| > \bar{A}$ | 0.5 | 92.15 | | Soft Only (Prune High) | $q_k \notin \mathcal{B}_t$ | 0.5 | 91.23 |
> | Hard Only (Prune Low) | $\|A\| \le \bar{A}$ | – | 92.35 | | Hard Only (Prune Low) | $q_k \in \mathcal{B}_t$ | – | 92.10 |
> | Hard Only (Prune High) | $\|A\| > \bar{A}$ | – | 90.96 | | Hard Only (Prune High) | $q_k \notin \mathcal{B}_t$ | – | 90.68 |
>
>
> ## **[W2] Limited Compatibility with Asynchronous RL**
>
> Thanks.  **DPPO targets standard synchronized group-based RL training (Guo et al., 2025; Zheng et al., 2025b; Lin et al., 2025)**. Asynchronous RL—motivated by heterogeneous latency in agentic or long-context settings—operates under a fundamentally different paradigm where synchronization is itself avoided. Extending DPPO to such settings is an interesting future direction, but outside the scope of this work, which we will clarify in the revision.
>
>
> ## **[Q1] Concerns About Stale Historical Statistics**
> Thanks. The asymmetry is intentional: completion pruning occurs post-rollout where current advantages are directly available, while prompt pruning must be pre-rollout to avoid the causality dilemma (line 224-228, Right). As shown in Table 2, the Pearson correlation between historical and current prompt scores increases steadily to 0.709 by epoch 14, confirming increasing reliability. The carry-forward mechanism (Eq. 16) and soft pruning (Eq. 17) jointly ensure average staleness remains well below 2 epochs throughout training (see Reviewer [1T6B], [Q3]).
>
>
>
> **Table 2. Pearson correlation between historical and current prompt scores (Qwen3-0.6B, $r_q = r_o = 0.5$).**
>
> | **Epoch** | 1 | 2 | 3 | 4 | 5 | 6 | 7 | 8 | 9 | 10 | 11 | 12 | 13 | 14 |
> |:---:|:---:|:---:|:---:|:---:|:---:|:---:|:---:|:---:|:---:|:---:|:---:|:---:|:---:|:---:|
> | **Correlation** | 0.454 | 0.476 | 0.543 | 0.599 | 0.605 | 0.629 | 0.643 | 0.656 | 0.673 | 0.677 | 0.684 | 0.691 | 0.704 | 0.709 |
>
>
> ## **[Q3] Gap Between Pruning Ratio and Actual Speedup**
> We thank the reviewer for this question. The pruning ratios $r_q$ and $r_o$ apply only to **candidate subsets**, not the full sample set: $r_o$ targets only low-$|A|$ completions (with high-$|A|$ completions always retained), and $r_q$ targets only the bottom 50% of prompts (Eq. 15 and Eq. 17). Consequently, the actual fraction of discarded samples at $r_q = r_o = 0.9$ is substantially smaller than a naïve "90%+90%" interpretation suggests, explaining why the realized speedup is well below a multiplicative upper bound.

---

> > ### Author Rebuttal · Reviewer_NSv6 · 2026-04-04
> >
> > Thank you for your detailed reply. While additional results on reversed pruning clarify one of my key concerns, the limitation on asynchronous RL is crucial yet completely omitted. I would like to see more discussions on this topic in additional responses.

---

> > > ### Author Response · Authors · 2026-04-04
> > >
> > > Thank you for the continued engagement. We provide a more detailed discussion here：
> > >
> > > Fully asynchronous RL frameworks inherently operate in an **off-policy regime**, trading on-policy correctness for throughput and introducing uncontrolled distributional shift that undermines gradient estimation integrity. In contrast, **DPPO accelerates training while strictly preserving on-policy correctness**. Directly migrating DPPO to fully asynchronous settings is therefore non-trivial: our importance rescaling and group-level pruning threshold both presuppose that all $G$ completions are drawn from the same behavioral policy $\pi_{\theta_{\text{old}}}$, an assumption violated under fully asynchronous execution.
> > >
> > > A principled extension, however, is achievable. The recently proposed **Periodic Asynchronous RL framework (Lu, 2026)** offers a compatible substrate by achieving asynchronous acceleration via a producer–consumer pipeline while provably preserving strict on-policy correctness — all rollouts within a batch share the same policy snapshot, and weight updates occur only after the full batch is consumed. This structural guarantee keeps the distributional assumptions of DPPO intact, allowing our hierarchical pruning and importance rescaling to be applied seamlessly once each complete group arrives in the training queue, thereby realizing efficient, unbiased policy optimization under asynchronous execution. We will incorporate this discussion into the revision and pursue empirical validation as a future direction.
> > >
> > > We also emphasize that DPPO targets the predominant synchronized group-based RL paradigm (Guo et al., 2025; Zheng et al., 2025b; Lin et al., 2025), which remains the standard setting for the vast majority of LLM reasoning works. Within this setting, DPPO delivers theoretically grounded acceleration with provably unbiased gradient estimation — a property that existing heuristic pruning methods (CPPO, GRESO) fundamentally lack — with consistent empirical gains across diverse models, datasets, and RL algorithms confirming its **practical significance**.
> > >
> > > We hope the above clarification adequately addresses the reviewer's concern. Given the theoretical rigor, broad empirical validation, and clear practical significance of DPPO within the standard synchronized RL paradigm — as well as the identified principled path toward asynchronous extension — we kindly invite the reviewer to reconsider the current score. Thanks, again.
> > >
> > > ### Reference
> > > Lu, J. Periodic asynchrony: An on-policy approach for accelerating LLM reinforcement learning.

---

### Decision · Program_Chairs · 2026-04-30

**Decision:**

Accept (regular)

**Comment:**

This paper proposes a method that enables dynamic pruning while preserving unbiased gradient estimation via importance sampling-based correction for GPRO acceleration. In addition, it introduces a packing method, a window-based greedy strategy that maximizes valid token density and hardware utilization to mitigate data sparsity issues. The paper was initially reviewed with two reviewers leaning toward acceptance and two toward rejection; after the rebuttal phase, the majority of reviewers shifted toward acceptance, with one reviewer remaining in the position. The AC has carefully reviewed the paper, all reviews, the rebuttal, and the subsequent reviewer discussions to ensure a fair and thorough assessment.

The main concerns raised by reviewers, including the need for additional experiments, questions regarding pruning parameters and performance variance, and presentation issues (e.g., clarity of formalization and potentially misleading descriptions), have been largely addressed during the rebuttal phase. Although Reviewer DR9x did not provide a rebuttal acknowledgment, the AC finds that their concerns appear to be substantially resolved. Reviewer NSv6 raised valid points, and the authors' follow-up responses, along with additional experiments, address most of these concerns, as the reviewer also acknowledged. The remaining concern, that the method may not be directly applicable to asynchronous RL settings such as asynchronous GRPO, is valid; however, the AC considers this to be beyond the intended scope of the paper.

Overall, the AC finds that the paper's strengths outweigh its weaknesses and recommends acceptance. The authors are encouraged to incorporate all additional experiments and clarifications provided during the rebuttal into the final version. The AC also encourages the authors to release code to improve reproducibility.